# Lower tropospheric ozone over the North China Plain: variability and trends revealed by IASI satellite observations for 2008-2016

Gaëlle Dufour[1], Maxim Eremenko[1], Matthias Beekmann[1], Juan Cuesta[1], Gilles Foret[1], Audrey Fortems-Cheiney[1], Mathieu Lachâtre[1], Weili Lin[2], Yi Liu[3], Xiaobin Xu[4], Yuli Zhang[3]

[1]Laboratoire Inter-universitaire des Systèmes Atmosphériques (LISA), UMR7583, Universités Paris-Est Créteil et Paris Diderot, CNRS, Créteil, France

[2] College of Life and Environmental Sciences, Minzu University of China,Beijing,China

[3] Institute of Atmospheric Physics, Chinese Academy of Sciences, Beijing, China

[4] Key Laboratory for Atmospheric Chemistry of China Meteorological Administration, Chinese Academy of Meteorological Sciences, Beijing, China

*Correspondence to*: Gaëlle Dufour (gaelle.dufour@lisa.u-pec.fr)

**Abstract.**

China, and especially the North China Plain (NCP), is a highly polluted region. Nevertheless, emission reductions have been occurred since about 10 years, starting with $SO_2$ emissions since 2006 and continuing with $NO_x$ emissions since 2010. Recent studies show a decrease in $NO_2$ tropospheric column since 2013 attributed to the $NO_x$ emissions reduction. Quantifying how these emission reductions translates to the ozone concentrations remains unclear due to apparent inconsistencies between surface and satellite observations. In this study, we use the lower tropospheric (LT) columns (surface-6km asl) derived from the IASI-A satellite instrument to describe the variability and trend of LT ozone over the NCP for 2008-2016. First, we investigate the IASI retrieval stability and robustness based on the influence of atmospheric conditions (thermal conditions, aerosol loading) and retrieval sensitivity changes. We compare IASI-A observations with the independent IASI-B instrument aboard the Metop-B satellite as well as surface and ozonesonde measurements. The conclusion of this evaluation is that the LT ozone columns retrieved from IASI-A are reliable to derive trend representative of the lower/free troposphere (3-5 km). Deseasonalized monthly timeseries of LT ozone show two distinct periods: a first period (2008-2012) with no significant trend (< -0.1 %/yr) and a second period (2013-2016) with a highly significant negative trend of -1.2 %/yr, leading to an overall significant trend of -0.77 %/yr for 2008-2016. We explore the dynamical and chemical factors that could explain these negative trends using a multivariate linear regression model and chemistry-transport model simulations to evaluate the sensitivity of ozone to $NO_x$ emissions reduction. The results show that the negative trend observed from IASI for 2013-2016 is almost equally attributed to large-scale dynamical processes and emissions reduction, the large El Nino event in 2015-2016 and the $NO_x$ emissions reduction being the main contributors. For the entire period 2008-2016, large-scale dynamical processes explain more than half of the observed trend, with a possible reduction of the stratosphere-to-troposphere exchanges being the main contribution. Large-scale transport and advection evaluated using CO as a proxy contributes for a small part of the trends (~10%). However, a residual significant negative

trend remains showing the limitation of linear regression models to account for non-linear processes such as ozone chemistry and stress the need of a detailed evaluation of changes in chemical regimes with the altitude.

## 1 Introduction

The rapid economic development and urbanization in China during the last three decades resulted in increasing pollutant emissions leading to the largest pollutant concentrations in the world, largely exceeding the recommended outdoor air pollutant thresholds from the World Health Organization (WHO). Several studies point toward a general ozone ($O_3$) increase over some parts of China mainly attributed to the emission increase to both surface (Cooper et al., 2014; Lu et al., 2018; Ma et al., 2016; Wang et al., 2009), as well as in the lower troposphere (Ding et al., 2008; Sun et al., 2016; Wang et al., 2017b), or in the entire troposphere (Chen et al., 2015; Verstraeten et al., 2015; Wang et al., 2012; Xu and Lin, 2011). Only a few long-term $O_3$ measurements are available in China. Wang et al. (2009) reported an increase of surface $O_3$ of 0.58 ppb/yr during 1994-2007 at a regional station in Hong Kong. Ding et al. (2008) derived an $O_3$ trend of 2%/yr between 1995 and 2005 in the lower troposphere from the MOZAIC commercial aircraft measurements. Xu and Lin (2011) analyzed tropospheric ozone trends from satellite using the TOR (tropospheric ozone residual) approach during 1979-2005 and found a trend of 1.10 DU per decade in summer over the North China Plain (NCP). More recently, Xu et al. (2016, 2018) reported on trends derived from surface measurements operated at Mt. Waliguan, on the Tibetan Plateau over the period 1994-2013. The derived general trend is about 0.1-0.3 ppbv/yr, with a more significant trend during spring and autumn, much smaller trend in winter and no significant trend in summer. Several studies are available on shorter time periods and for more recent years. In Beijing, Tang et al. (2009) reported on ozone trends of 1.1ppb/yr for the 2001-2006 period. Ma et al. (2016) and Sun et al. (2016) found significant increase of surface ozone at two stations representative of the NCP for the 2003-2015 period. Their analyses showed a trend of 1.13 ppb/yr at Shangdianzi and a trend of 2.1 ppb/yr during summertime at Mt. Tai, respectively. Verstraeten et al. (2015) show, using TES satellite observations, that tropospheric ozone over China has increased by about 7% between 2005 and 2010, because of the rise in Chinese emissions and an increase in the downward transport of stratospheric ozone (Neu et al., 2014). Most of the long-term trends are attributed to the large increase of precursor emissions, such as the $NO_x$ emissions, which have tripled since 1990 (e.g. Lin et al., 2017; Richter et al., 2005). However, ozone concentrations are also influenced by other factors, in particular dynamical factors, which drive most of the variability of ozone (e.g. Wespes et al., 2017b) with potential modulations of the trends. Among the processes impacting ozone concentrations, stratosphere-to-troposphere transport that brings ozone-rich air down to the surface in some cases (e.g. Dufour et al., 2010, 2015; Lin et al., 2015; Verstraeten et al., 2015) is one key parameter as processes that modify the large-scale atmospheric circulation such as the El Nino-Southern Oscillation (ENSO), the quasi-biennial oscillation (QBO), the solar cycle (e.g. Ebojie et al., 2016; Oman et al., 2013; Wespes et al., 2016, 2017b). Facing the large pollutant increase since the 90s, China has started implementing stringent air quality controls starting in 2006 with reductions of $SO_2$ emissions and followed by successful emission reductions of $NO_x$ more recently (e.g. van der A et al., 2017; Li et al., 2017; Ma et al.,

2016). Only few studies evaluate the recent trends of ozone concentrations for period encompassing the recent changes in $NO_x$ emissions. Ma et al. (2016) used ozone data collected at the Shangdianzi background station, representative of the NCP to derive trends over 2003-2015. They did not find any significant correlation between ozone and $NO_2$ trends. They state that the changes of VOC emissions and the ratio $VOC/NO_x$ might play a more important role in the observed increase of ozone

than the reduced NO titration induced by $NO_x$ emission reductions in agreement with conclusions of Sun et al. (2016) based on measurement at the Mt. Tai station. A very recent work, based on the China National Environmental Monitoring Center (CNEMC) network, also points toward an increase of surface ozone in response to the NOx emissions reduction in VOCs-limited regions (Lu et al., 2018). Another very recent work done in the framework of the Tropospheric Ozone Assessment Report supported by the IGAC (International Global Atmospheric Chemistry) community states that ozone is generally

increasing at the global scale over the recent decade. However, some inconsistencies have been reported between infrared (IR) sounders like IASI and ultraviolet (UV) sounders like OMI. Trends derived from IR sounders are mainly negative, whereas they are positive when derived from UV sounders (Gaudel et al., 2018). One hypothesis to explain this discrepancy relies on the difference in vertical sensitivity. A particular attention on the instrumental and retrieval stability has then to be paid when using satellite observations to derived tropospheric ozone trends.

In this study, we focus our analysis over the lower troposphere (LT) of the NCP using the thermal infrared IASI satellite observation for 2008-2016. We analyze the variability and recent trend (2008-2016) of the LT ozone columns with respect to different dynamical factors and proxies to account for emissions changes, such as $NO_2$ and HCHO tropospheric columns and carbon monoxide columns. Section 2 describes the IASI ozone observations, the method to calculated the trends, and the developed multivariate linear regression model. Section 3 evaluates the instrumental and retrieval stability of IASI and

discusses the reliability of the IASI derived trends. Section 4 provides an analysis of the variability and trends of LT ozone over the NCP over nine years (2008-2016) based on the IASI instrument onboard the Metop-A satellite, operational since 2006. Conclusions are given in Section 5.

## 2 Satellite data and method

### 2.1 IASI satellite data

The IASI (Infrared Atmospheric Sounding Interferometer) (Clerbaux et al., 2009) instruments are nadir-viewing Fourier transform spectrometers. They have been planned for flying on board the EUMETSAT (European Organisation for the Exploitation of Meteorological Satellites) Metop satellite. Currently, two versions of the instrument are operational: one aboard the Metop-A platform since October 2006 and the Metop-B platform since September 2012. The IASI instruments operate in the thermal infrared between 645 and 2760 $cm^{-1}$ with an apodized resolution of 0.5 $cm^{-1}$. The field of view of the

instrument is composed of a $2 \times 2$ matrix of pixels with a diameter at nadir of 12 km each. IASI scans the atmosphere with a swath width of 2200 km and crosses the equator at two fixed local solar times 9:30 am (descending mode) and 9:30 pm (ascending mode), allowing the monitoring of atmospheric composition twice a day at any location. The two Metop satellites

are on the same orbit shifted by 180° leading to a time difference of about 50 minutes between the two IASI instruments (Boynard et al., 2018).

Ozone profiles are retrieved from the IASI radiances following the method described in Eremenko et al. (2008) and Dufour et al. (2012, 2015). The retrieval algorithm is based on the KOPRA radiative transfer model and its inversion tool (KOPRAFIT). A constrained least squares fit method with an analytical altitude-dependent regularization is used. The regularization matrix is a combination of first order Tikhonov constraints (Tikhonov, 1963) with altitude-dependent coefficients (Kulawik et al., 2006). The coefficients are optimized both to maximize the degrees of freedom (DOF) of the retrieval and to minimize the total error on the retrieved profile. Different a priori and constraints are used depending on the tropopause height, which is calculated from the temperature profile retrieved from IASI using the definition based on the lapse rate criterion (WMO, 1957). Three situations are considered: polar (<10 km), midlatitudes (10-14 km), and tropical (>14km). The a priori profiles are compiled from the ozonesonde climatology of McPeters et al. (2007). As shown in Dufour et al. ( 2010, 2012), two semi-independent partial columns of ozone can be considered between the surface and 12 km: the lower-tropospheric column integrating the ozone profile from the surface to 6 km altitude – above sea level (asl) – and the upper-tropospheric column integrating the ozone profile from 6 to 12 km altitude.  Note that the latter column can include stratospheric air masses depending on the tropopause height. The averaging kernels give information on the vertical sensitivity and resolution of the retrieval. The lower tropospheric column shows a maximum sensitivity typically between 3 and 4 km with a limited sensitivity to the surface (Dufour et al., 2012). From the retrieved profiles, different ozone partial columns can be calculated. The lower tropospheric column (LT) from the surface up to 6 km (asl) is considered in this study. Note that only the morning overpasses of IASI are considered for this study in order to remain in thermal conditions with a better sensitivity to the lower troposphere.

Recent studies based on IASI observations, mainly ammonia, reported on changes in the temperature product delivered by EUMETSAT that impact the retrieval. The changes are related to different versions of the product (Van Damme et al., 2017). In order to avoid the potential impact of versioning of the auxiliary parameters (such as temperature profile, clouds screening, etc) on the ozone retrieval, we apply a self-consistent procedure. Surface temperature and temperature profiles are retrieved before the ozone retrieval. A data screening procedure is applied to filter cloudy scenes and to insure the data quality (Dufour et al., 2010, 2012; Eremenko et al., 2008).

### 2.2 Timeseries analysis method

The IASI observations are analyzed over a nine-year period (2008-2016) for IASI-A, the first instrument aboard Metop-A satellite and over a four-year period (2013-2016) for IASI-B, the second instrument aboard the Metop-B satellite. Each pixel is retrieved individually and filtered following the previously described quality flags. Gridded monthly averages are computed on a 0.25°x0.25° resolution grid from daily averages for the East Asia domain (20-48°N, 100-150°E). As the principal focus of this study is the North China Plain, we then calculate regional averages from the gridded monthly means for this region to derive the monthly timeseries over the NCP. The domain considered for the NCP ranges between 35°N and

41°N in latitude and between 114°E and 122°E in longitude (Fig. 1). Seasonal and annual timeseries are derived from the regional monthly timeseries.

We also calculate the deseasonalized monthly timeseries using the average-percentage method. A climatological index, $sindex$, is calculated over the considered nine-year period following Eq. (1).

$$sindex(im) = \frac{1}{9}\sum_{iy=1}^{9}\frac{month\_ave(im,iy)}{year\_ave(iy)} \qquad (1)$$

where $month\_ave$ is the monthly average for the NCP calculated as described previously for each month ($im$) and each year ($iy$), and $year\_ave$ is the yearly average. The climatological index is then applied to the monthly timeseries to remove the

seasonal component from the series and obtain the deseasonalized timeseries $deseas$ (Eq. (2)).

$$deseas(im,iy) = \frac{month\_ave(im,iy)}{sindex(im)} \qquad (2)$$

In the following, the Theil-Sen estimator (Sen, 1968) and the nonparametric Mann-Kendall test (Kendall, 1975) are used to

estimate the linear trend magnitude and to determine the significance of the trends (95% confidence range), following the recommendation of the TOAR (Lefohn et al., 2018). All the linear trends presented in the current study are computed based on the deseasonalized timeseries. We also calculate the anomalies against the mean over the entire period.

**2.3 Regression model**

In order to evaluate the main processes contributing to the trends derived in the current study, we developed a multivariate

linear regression model. Multivariate linear regression methods have been extensively used to determine the processes driving the variability and trends of stratospheric (e.g. Oman et al., 2010a, 2010b; Stolarski et al., 2006) and tropospheric (e.g. Ebojie et al., 2015; Wespes et al., 2016, 2017a) ozone. We apply the regression on the deseasonalized monthly timeseries discussed previously, following Eq. (3).

$$O_3(im) = b + t.im + \sum_j m_j X_j(im) + \varepsilon(im) \qquad (3)$$

where $O_3$ is the deseasonalized monthly mean LT ozone, $im$ is the month index (starting in January 2008), $b$ is the intercept, $t$ is the slope from which the trend is calculated, and $\varepsilon$ is the error term. The $X_j$ are the different normalized explicative variables considered in the fit with $m_j$ the fitting coefficients. The explicative variables are normalized over the 2008-2016

period. The significance of including or excluding a variable is evaluated using the p-value. We consider a range of 95% confidence (i.e. $p<0.05$). Each variable was tested individually first and combined then with the other variables. The variables that were not significant were removed in the final fit. Note that the regression model has been developed using

predefined functionalities of Python/Pandas programing tools, which does not include the Theil-Sen estimator. However, we checked that the linear trend derived from the regression model is not significantly different from those derived using the Theil-Sen estimator. We have tested different dynamical variables similar to (Wespes et al., 2017b) in the regression model, related to the solar activity, the dynamical processes leading to a modulation of the stratospheric circulation and of the stratospheric-tropospheric exchanges (STE) and influencing the tropospheric ozone such as the quasi-biennale oscillation (QBO), the El Nino/Southern Oscillation (ENSO). The tested variables are:

- the 10.7 cm solar radio flux. The monthly means have been calculated from daily data, taken from the NOAA National Weather Service Climate Prediction center: ftp://ftp.ngdc.noaa.gov/STP/space-weather/solar-data/solar-features/solar-radio/noontime-flux/penticton/penticton_adjusted/listings/listing_drao_noontime-flux-adjusted_daily.txt (February 2018).

- The QBO at 10 and 30 hPa are considered and summed up. They are taken from http://www.geo.fu-berlin.de/met/ag/strat/produkte/qbo/singapore.dat (February 2018).

- The Multivariate ENSO index (MEI), taken from https://www.esrl.noaa.gov/psd/enso/mei/table.html (February 2018).

- The tropopause height, given by the geopotential height for 2 PVU and the potential vorticity at 300 hPa. The data are taken from the ERA-interim reanalysis at http://apps.ecmwf.int/datasets/data/interim-full-daily/ (February 2018).

In addition to these dynamical variables, some chemistry-related variables used as proxies for emission changes in the NCP have also been tested. In particular, we use the tropospheric $NO_2$ columns derived from OMI (Boersma et al., 2007, available from TEMIS database www.temis.nl), the tropospheric HCHO columns derived from OMI (De Smedt et al., 2015, 2018, available from TEMIS database www.temis.nl) and the total CO columns derived from IASI (George et al., 2009, available from AERIS database www.aeris-data.fr).

## 3 Evaluation of the reliability of the IASI derived trends

### 3.1 Retrieval stability

In this section, we evaluate the different factors that could impact the stability of the retrieval during the 2008-2016 period over the NCP and then the reliability of the trends derived from IASI ozone observations. We consider the atmospheric conditions that could influence the retrieval and analyze the timerseries of related parameters. The thermal infrared measurements such as those from IASI are very sensitive to the thermal conditions of the measured scene. The surface temperature and the thermal contrast are two parameters that drive the sensitivity of the thermal infrared measurements. They can be derived from the IASI observations themselves. However, in order to be independent of the IASI observations and of a possible change in the instrumental stability, we consider the skin temperature and the temperature at 2 meters from the ECMWF reanalysis (ERAinterim) to evaluate the variability and trend of these parameters over the NCP and during the

2008-2016 period. The trends derived from the deseasonalized timeseries are not significant with p-values of 0.08 and 0.32, larger than 0.05, for the skin temperature and the thermal contrast (calculated from the skin temperature and the 2-meter temperature), respectively. However, timeseries of monthly skin temperature show a singular change starting at the end of 2013 with temperature larger especially during wintertime (Fig. 2a). The thermal contrast timeseries do not reveal such a

change (not shown). We also checked that the thermal contrast calculated directly from the IASI observations does not show trends and changes in 2013 (Fig. 2b).

The other parameter, which may influence the retrieval, is the tropopause height and its possible evolution during the considered period. Indeed, as mentioned in section 2, different constraints and a priori profiles are used depending on the tropopause height. Trends in the tropopause height may then influence the retrieval. Moreover, depending on the depth of the

troposphere, the LT ozone column calculated up to a fixed altitude (6 km) is more or less influenced by upper tropospheric and lower stratospheric air. We consider both the tropopause height derived from the IASI temperature profiles (lapse rate method) and the tropopause given by the 2 PVU geopotential given by the ERAinterim reanalysis to evaluate the evolution of the tropopause height during the 2008-2016 period. Both datasets lead to similar monthly timeseries with a calculated trend of 0.02 km/yr, but not significant since $p > 0.05$ (p=0.32 and p=0.15 respectively).

Another atmospheric condition that may influence the ozone retrievals of IASI is the presence of (coarse) aerosols. Indeed, aerosols have broad spectral signature in the spectral region used for the ozone retrieval. China is known for experiencing large aerosol loading that may affect ozone retrieval. Usually, we assume that the retrieval quality filters allow one to reject the most affected situations. However, in order to evaluate the potential impact of aerosol loading on the retrieval and then on the derived trend, we filter out the IASI observations when the aerosol optical depth (AOD) measured by MODIS ((Hsu

et al., 2013; Levy et al., 2013), https://giovanni.gsfc.nasa.gov) are larger than 0.2. Figure 2e shows the monthly timeseries and the derived linear trends. The calculated linear trend (-0.19±0.04 DU/yr) is similar to the trend derived for data without aerosol filters (see section 4), and significant ($p < 10^{-3}$).

In addition, we based our evaluation of the retrieval stability to the analysis of the averaging kernels (AK). Indeed, they integrate and translate the retrieval sensitivity to the atmospheric conditions (temperature, pressure, etc changes). We

consider two related variables: the degrees of freedom (DOF) of the retrieval, calculated as the trace of the AK matrix, and the altitude of the maximum sensitivity of the retrieval. The DOF and the altitude of maximum sensitivity are calculated for the LT ozone column. The resulting monthly timeseries averaged over the NCP are displayed in Fig. 2c-d. The linear trends derived for these two variables are increasing (0.002 per year) for the DOF and decreasing (-0.02 km/yr) for the altitude of maximum sensitivity but not significantly since p>0.05 (p=0.06 and p=0.12 respectively).

Finally, we made the following numerical experiment. We consider an atmospheric situation where ozone vertical distribution would be constant over the NCP for the entire 2008-2016 period, leading to no-trend situation. The unique ozone profile and the associated pressure and temperature profiles are taken from a chemistry-transport model – here the LMDz-INCA model (Hauglustaine et al., 2004). We apply the actual AK of each individual IASI pixel retrieved between 2008 and 2016 to this unique profile and calculate the resulting ozone LT columns. The deseasonalized timeseries is then used to

evaluate the resulting trend (Fig. 3). Note that the observed variations indicates the changes in the meteorology, surface conditions, etc, that influence the retrieval sensitivity (and then the averaging kernels) along the year. Despite these variations, the linear trend calculated from the deseasonalized timerseries is negligible (0.005 DU/yr) and not significant since $p > 0.05$ ($p = 0.79$). Thus, we can conclude from this experiment that no significant trend can be attributed to a change in the retrieval sensitivity for the NCP during the 2008-2016 period.

## 3.2 Comparison with the independent IASI-B ozone observations

The trends derived in this study are computed from the IASI-A instrument since it covers the entire 2008-2016 period. Since February 2013, the second IASI instrument aboard the Metop-B satellite has been also providing data. In this section, we use the IASI-B instrument for the period 2013-2016 and compare the monthly timeseries and trends to those derived from the IASI-A instrument. For the comparison of the two instruments, the monthly averages are calculated from daily (morning overpass) gridded data at a resolution of 0.25°. The grid cells considered in the average are those for which data are available for the two instruments. Figure 4 shows the results of the comparison. A positive bias of +0.41 DU (+2%) is observed on average between IASI-B and IASI-A. This is in agreement with the results, obtained with a different retrieval algorithm, reported by Boynard et al. (2016, 2018) for tropospheric ozone. The trends derived from IASI-A and IASI-B from deseasonalized timeseries (Fig. 4) are very similar, -0.33±0.05 DU/yr and -0.32±0.06 DU/yr, respectively. For comparison, the trend derived for the same period with all the IASI-A data considered, not only those in coincidence with IASI-B, is -0.24±0.06 DU/yr. Thus, the comparison of IASI-A and IASI-B confirms that the trend derived from IASI-A for the period 2013-2016 is not due to an instrumental drift or an instrumental failure of IASI-A as the IASI-B instrument provides independent measurements.

## 3.3 Comparison with ozonesonde measurements

We also performed a validation by comparing the IASI observations with ozonesonde measurements available in the East Asian region. We use the same method for the comparison than those described in Dufour et al. (2012, 2015). We compare IASI ozone columns to the ozonesonde columns smoothed with the IASI averaging kernels. Five ozonesonde stations are used for the validation. They are listed, as well as the obtained results, in Table 1. The covered time period extends from 2008 to 2015 (at the time of the study, the sonde data were not available for the entire year 2016 for all the sondes). The coincidence criteria used for the present validation exercise are 1° around the station, a time difference smaller than 12h, and a minimum of 10 cloud-free pixels matching the two previous criteria. The criterion on the time difference has been relaxed compared to a previous study (Dufour et al., 2015) in order to have a more statistically-significant number of coincidences for all the stations.

The bias between IASI observations and ozonesondes measurements is negative and ranges from -3% (Sapporo) to -26% (Beijing). It is worth noting that the instrumentation of the Beijing ozonesonde has changed in 2013. Comparisons with IASI before 2013 show a negative bias of about -26%, whereas the bias decreases to -11% in 2014, being in better agreement with

other Asian sondes (Zhang et al., 2014). On average, the bias for the Asian stations is about -10 to -15%, IASI underestimating the LT ozone columns. We also compare the results for the first four years of the period and the last four years of the period (Table 1). We observe a degradation of the comparison results between IASI and the sonde at the beginning and the end of the period. For example, the negative bias increases from 10% to 14% at the Tateno station and the correlation coefficient decreases from 0.87 to 0.75. The number of days with coincident measurements is not that high (about 20 to 25 per year per sonde) for the Asian stations. It may introduce a sampling issue, which could explain this difference. To demonstrate this, we choose the midlatitude European station with the largest number of available measurements, the Payerne station to do another comparison. In that case, a small bias of +2.8% is observed but with a poor correlation (Table 1). We also observed a significant change in the bias from the beginning to the end of the period. Looking at the results in details, it arises that the winter period was not sampled (no coincidence) for three years over four during the beginning of the period (2008-2011). Filtering out the winter period (DJF) in the comparison IASI/sonde leads to a much better agreement with a small bias (-0.12%) and a good correlation (r=0.68), and no significant degradation of the comparison between the beginning and the end of the period (Table 1, last row). However, removing winter season in the comparison for the Asian sondes does not improve the comparison, except slightly for Sapporo. We also compared IASI-B and IASI-A LT ozone columns to the ozonesonde measurements for the 2013-2016 period, using only days for which observations are available for the three datasets. We obtain very similar results. For example, the bias for the Tateno soundings is the same (-15%) and the correlation coefficient is slightly improved with IASI-B (0.71 against 0.65).

Despite the poor temporal sampling frequency of the ozonesondes (at the best about four per month for Asian sondes), we calculate the slope of the seasonal timeseries for the IASI and smoothed ozonesonde LT columns for each station, as a first approximation of the trend (Table 1). Almost all of the slopes are not statistically significant. This is clearly visible with the standard deviations larger than the slopes themselves. The only soundings for which the slope is (slightly) significant are the Payerne station with a good agreement when winter measurements are not considered (see discussion above), the Tateno and the Sapporo stations with a poor agreement. Figure 5 compares the annual variations of IASI LT columns and sondes LT columns, both without and with averaging kernels applied for the four Asian ozonesondes. Whereas IASI exhibits rather small interannual variability and relatively flat timeseries, the ozonesondes, especially in the Tateno and Sapporo stations, exhibit an increase in 2010-2011 with stabilization the following years. This is clearly visible on the raw sounding (i.e. without averaging kernel smoothing). One possible explanation for this increase is the change of the used technology for the sondes. The sounding technology has moved from KC-96 sondes to ECC sondes in December 2009 for Tateno and Sapporo soundings (Morris et al., 2013). This increase translates into the slopes we can calculate from ozonesondes leading to positive slopes (Table 1). In order to test the sensitivity of the derived slope, especially its sign, to the number of samples, we use the Tateno station during the IASI-B period during which no instrumental change was made on the sondes. We calculate the slope for the Tateno ozonesondes in two situations: (i) when IASI-A, IASI-B, and the sondes match the coincindence criteria (30 days sampled), (ii) when IASI-B only and the sondes match the concidence criteria (48 days sampled). The slopes we obtain are: -0.03±0.2 (DU) and 0.26 ±0.2 (DU), respectively for the sonde measurements and -0.31±0.2 (DU) and

0.15±0.2 (DU), respectively for IASI-B. As expected in the case of poor sampling as for the sondes, changing the number of samples can completely change the slope of the linear regression and change the sign of the slopes in this particular case. This combined with the instrumentation changes for some stations (Beijing, Tateno, Sapporo) stresses the limitation of using ozonesondes to evaluate the trends derived from satellite observations in our case.

## 4 Variability and trends of LT ozone over the NCP

### 4.1 Variability and trends derived from IASI-A: 2008-2016

Figure 6a shows the monthly timeseries of the LT ozone column from January 2008 to December 2016 over the NCP. A large seasonal cycle with an average amplitude of about 5.7 DU is observed with a maximum observed mainly in June and a minimum observed in December/January as already reported (e.g. Ding et al., 2008; Dufour et al., 2010; Hayashida et al., 2015; Safieddine et al., 2016). The interannual variability is small, about 0.15 DU (< 1%), in the first five years. A drop of 0.74 DU is observed in 2013, followed by successive decreases in 2015 and 2016 (Fig. 6b). These decreases are also seen in the anomalies. The anomaly is negative during the first half of the year in 2013 and 2014 and all over the years 2015 and 2016 (Fig. 6c). Seasonal analyses of the timeseries suggest that the ozone drop observed in 2013 is mainly driven by the decrease of 1.5 DU observed in spring (MAM – March, April, May) the same year (Fig. 6d). For the other seasons, the behaviors are different but also contribute partly to the interannual variations and the significant decrease observed since 2013. LT ozone does not exhibit significant variations during the SON (September-October-November) period, except in 2015, where a larger decrease is observed, likely contributing to the decrease of ozone observed at the end of the period in the annual and monthly timeseries. The winter period (DJF – December, January and February) is marked by a decrease of about 2 DU between 2008 and 2013, followed by a slight increase the following years. During the summer period (JJA – June, July, August), the LT ozone increases from 2008 to 2011 (+1.2 DU) and starts a continuous decrease (except in 2014) of about -1.8 DU from 2011 to 2016. Finally, we calculate the trends from the deseasonnalized timeseries (Fig. 6e). For the entire period, the trend is negative (-0.17±0.02 DU/yr, -0.774±0.001 %/yr) and significant (p<0.05). We also calculate the trend over the two distinct periods identified from the annual and seasonal evolution of LT ozone: 2008-2012 and 2013-2016. No significant trend (with slope close to zero, -0.02±0.05 DU/yr ) is obtained for the first period and a significant negative trend of -0.24±0.06 DU/yr (-1.161±0.003 %/yr) is obtained for the second one. As already mentioned, similar negative trends have been reported from IASI tropospheric ozone columns in the Northern Hemisphere (Wespes et al., 2016, 2017a) with some inconsistencies with other satellite observations and in situ measurements (Gaudel et al., 2018). The conclusions of Section 3 show that no retrieval drift or instrumental instability has been noticed that could explain the observed trend. It is worth noting that the trend derived from the IASI-B instrument for this second period is in agreement with the one reported here (see Section 3). In order to evaluate if the LT columns can be strongly contaminated by the altitudes higher in the troposphere and the stratosphere, we also derive the trends for different partial columns: one ranging from 6 to 12 km and considered as the upper tropospheric column (UT), the tropospheric ozone column (TOC) ranging from

the surface up to the tropopause, and the total column. Note that the UT column can include part of the lower stratosphere when the tropopause is lower than 12 km. The deseasonalized timeseries are plotted in Fig. 7 with the derived trends indicated in the figure. The UT and total columns do not show any trend whereas the TOC column presents a significant negative trend, likely driven by the negative trend observed in the lower troposphere. These results show that the negative trend observed in the lower troposphere with IASI is likely representative of the ozone evolution within the lower and more likely the free troposphere (3-5 km) where the IASI retrieval is the most sensitive. In the following, we explore the processes that could explain the ozone decrease observed in the LT from IASI in the NCP region. Several concomitant processes could explain the observed variability and trends: (i) recent studies show that several dynamical processes such as the QBO (Quasi-Biennale Oscillation), ENSO (El Nino Southern Oscillation), STE (Stratospheric-Tropospheric Exchange), etc influence the variability and trends of the tropospheric ozone column (e.g. Ebojie et al., 2015; Heue et al., 2016; Oman et al., 2013; Wespes et al., 2016, 2017a, 2017b). The long-range transport of ozone or precursors could also influence regional LT ozone by advection; (ii) Intensive emission regulations have been applied in China to reduce $SO_2$ and $NO_x$ emissions during the last years (van der A et al., 2017; Li et al., 2017). The emission reduction of $NO_x$, which are ozone precursors, is observed in the satellite $NO_2$ columns since 2013 as shown in Fig. 8 and reported in very recent inventories (Zheng et al., 2018). On the contrary, the emissions of anthropogenic volatile organic compounds (VOCs) are not regulated and do not show any decrease in the recent years. Zheng et al. (2018) report on an increase between 2010 and 2014 and stagnation since 2014. Stavrakou et al. (Stavrakou et al., 2017) report on an increase in 2013 and 2014 compared to previous years from OMI-HCHO-based emissions and attribute it to the economic recovery after the 2008-2009 crisis. Looking at the timeseries of the HCHO tropospheric columns derived from OMI (De Smedt et al., 2015), available from the TEMIS database, a continuous increase is well observed starting in 2013 and extending to 2016 (Fig. 8). It is worth noting that the increase is less observable in a more recent version of the HCHO product, except for the last year (De Smedt et al., 2018). Thus, one hypothesis is that reductions in surface emissions of $NO_x$ and increase or stagnation in VOCs emissions might cause a decreasing trend in lower tropospheric ozone levels as observed with IASI (Figs. 7-8).

## 4.2 Role of $NO_x$ emission reduction

In order to evaluate the impact of emission reduction on ozone, we use first the surface measurement done at the Shangdianzi station, China. The Shangdianzi station is a regional Global Atmosphere Watch (GAW) station, located about 100 km northeast of Beijing and classified as a rural station. Previous studies suggest that the pollutant observations at this station are representative of the regional-scale air quality of the NCP (Lin et al., 2008; Xu et al., 2009) and then more comparable to satellite observations than urban stations. Recent studies show a positive trend for surface ozone levels in the NCP (Ma et al., 2016; Sun et al., 2016). The timeseries are plotted in Fig. 9a-b for the 2009-2015 period. The calculated trend based on the deseasonalized timeseries is positive 0.31±0.18 ppb/yr or 0.80±0.46 %/yr. However, the trend is only slightly significant over this time period since p=0.09. We compare the surface measurements to the IASI LT columns, converted into equivalent volume mixing ratios. IASI observations within 0.25°x0.25° around the stations are considered

(40.5°N-40.75°, 117°E-117.25°E). Considering all the surface data (daily, hourly), the linear trend calculated from the deseasonalized timeseries for the surface station (Fig. 9a-b) is positive (slightly significant, p=0.09), whereas the IASI trend (calculated only for clear-sky days) is significantly negative. Obviously, the quantities are not completely comparable as we compare columns and surface measurements and as IASI is poorly sensitive to the surface (Cuesta et al., 2018). IASI

observations are made during the day and are more representative of the free troposphere or of highly developed planetary boundary layers (PBL) (Eremenko et al., 2008) and cannot be compared to nighttime observations when the PBL is isolated from the free troposphere. Then, we consider only daytime (8-20h local time) surface observations, which should be more representative of IASI observations. The calculated linear trend of the daytime surface measurement is still positive, but reduces and becomes poorly significant with p=0.74 (Fig. 9c-d). Then, we consider daytime surface observations only on the

days for which IASI data are available. The calculated linear trend becomes negative (not significant, Fig. 9f). We also consider the surface measurement the day after the IASI data are available. Indeed, a recent study shows that the downward mixing of free tropospheric ozone may largely impact the morning level of ozone in the surface layer, the surface ozone level on one day being likely related with ozone at higher levels the day before (Wang et al., 2017a). In that case, the calculated trend is even more negative but still poorly significant (Fig. 9h). These results illustrate the sensitivity of the trend

calculation to the sampling (day/night, clear-sky conditions) and stress the need to compare datasets with different temporal sampling frequency over subsets of data with consistent sampling before drawing conclusions. Beyond that, the answer given by the surface and satellite observations are inconsistent with surface measurements showing positive and/or not significant trends and satellite observations a significant negative trend. If both trends are reliable, a possible explanation to this inconsistency may be that the LT and surface ozone respond differently to the recent reduction of $NO_x$. Previous ozone

production efficiency studies (Ge et al., 2010, 2012) suggest that even at the background site of the NCP, ozone production in the surface layer seems to be more VOC-limited. Although Chinese $NO_x$ emissions have been reduced in recent years, the VOCs emissions have been increasing or stagnating as mentioned previously. The observed recent decline of tropospheric $NO_2$ (Fig. 8) might have contributed mainly to the decrease of ozone at the levels above the surface layer, where ozone production is more sensitive to $NO_x$. To better understand the changes of ozone at different altitudes over the NCP, we use

simulation experiments of the chemistry-transport model CHIMERE (Menut et al., 2013) made in the framework of another study (Lachâtre et al., in prep[1]). Two runs of CHIMERE with different emissions are compared for the year 2015. The first one was performed based on the EDGAR-HTAP-v2.2 2010 emission inventory (Janssens-Maenhout et al., 2015) and considered as the reference case. For the second run, the $SO_2$ and $NO_2$ OMI tropospheric columns were used to update the $SO_2$ and $NO_x$ emissions using a simple mass-balance method for the emission correction. The corrected emissions include

then the reduction of $NO_x$ emissions occurring the last years, which is about 25% over the NCP. Note that the VOCs emissions are the same for the two simulations. Figure 10 shows the differences (%) between the annual mean ozone concentration simulated with updated emissions and with the reference case at the surface level and at ~4km. At the surface,

---

[1] Lachatre, M., Fortems-Cheiney, A., Beekmann, M., Foret, G., Dufour, G., and Siour, G: The unintended consequence of $SO_2$ and $NO_2$ regulations over China: increase of ammonia levels and impact on $PM_{2.5}$ concentrations, in prep.

the ozone concentrations simulated with reduced $NO_x$ emissions are larger by 13% on average over the NCP. This corroborates the reported ozone increase, associated to the $NO_x$ emission reduction. On the contrary, the ozone concentrations at 4 km decrease compared to the reference when $NO_x$ emissions are reduced. The impact is small, -0.25 % on average over the NCP but it persists in the altitude range between 3 and 7 km, the range where the IASI observations are the most sensitive. These results suggest that our hypothesis concerning the response of LT or free tropospheric ozone (decrease) to the $NO_x$ reduction is credible and likely associated to the chemical regime changing from VOC-limited in the boundary layer to $NO_x$-limited in the free troposphere. Quantifying the change of chemical regime with the altitude is out of the scope of this study and would require observations with a better vertical resolution than those offered by satellite observations such as those from the IAGOS program (Petzold et al., 2015) and detailed model studies. The changes due to the $NO_x$ emission reduction on free tropospheric ozone remain small (Fig. 10) and do not allow to explain by themselves the negative trend observed with IASI. In the next section, we explore the processes contributing to the ozone variability and trend in addition to the $NO_x$ emission reduction.

### 4.3 Explicative variables

The multivariate linear regression model described in Section 2 was used to determine the explicative variables of the negative trend observed by IASI in the LT. The model has been applied for the entire 2008-2016 period but also for the 2013-2016 period for which the negative trend is the most significant. As the study focuses on the trend explanation, we applied the model to 3-month moving-average (deseasonalized) timeseries, the month-to-month variations being possibly more affected by the IASI sampling limited to clear-sky conditions. After a first application of the regression model to determine the significant explicative variables as described in Section 2, we applied the model introducing the variables one by one. In order to evaluate the fit, we calculate (using the Theil-Sen estimator) the trend of fit residual, named residual trend in the following, as well as the standard deviation of the residual. The results are reported in Table 2 for the two studied time periods.

After the fitting procedure, the significant variables for the 2008-2016 period are: the QBO, the potential vorticity (PV) at 300 hPa, the ENSO index, and the CO total columns deseasonalized timeseries derived from IASI. The normalized timeseries are displayed in Figure 11. The QBO account mainly for the low frequency variations and with a high significance ($p<10^{-3}$), but it only explains 12% of the initial trend (Table 2). The potential vorticity (PV) at 300 hPa has been chosen in order to account for the impact of stratospheric/tropospheric exchanges on the LT ozone. A significant decrease and trend are observed in the PV timeseries for the 2008-2016 period (Fig. 11), it allows one to explain 24% of the decreasing trend observed with IASI in the LT ozone. The normalized ENSO index shows an increase over 2008-2016 with two specific periods corresponding to two strong El Nino events in 2009-2010 and in 2015-2016. Introducing the ENSO index in the fit allows one to explain 18% of the initial trend (Table 2). The last significant variable, we found for this period, is the monthly CO. The CO variable, due to its long lifetime, is considered as a proxy for large-scale emission changes that may affect LT ozone regionally. To account for the long-range transport and advection, the regression model has been tested either with the

CO timeseries averaged for the Northern Hemisphere or with the CO timeseries averaged only over the NCP. Considering the CO averaged over the NCP leads to larger reduction of the residual trend (12% against 6%) suggesting that regional and hemispheric changes can both influence slightly the LT ozone trend. During the 2008-2016 period, the dynamical processes such as the QBO, the ENSO, and the STE seem to be the main drivers of the trend observed with IASI. The regional and hemispheric emission changes approximated by CO contribute only slightly (12%). Among the significant explicative variables, the PV, showing also a decrease during 2008-2016, is the explicative variable contributing the most to the calculated linear trend. This points toward a possible reduction of the stratospheric/tropospheric exchanges leading to a reduction of ozone levels in the free/lower troposphere. However, the different explicative variables explain only 65% of the observed trend. The residual trend remains significant and as high as -0.06 DU/yr. An additional trend is necessary to include in the fit in order to come up with a negligible not significant residual trend (0.005 DU/yr) (Table 2). Thus, explicative variables are still missing to fully explain the decreasing trend observed with IASI. One hypothesis, we can formulate, concerns the limitation of the linear regression model to account for nonlinear processes such as the chemistry driving the ozone production.

Concerning the 2013-2016 period, the significant variables are: the QBO, the PV at 300 hPa, the ENSO index, the $NO_2$ tropospheric columns and the HCHO tropospheric columns. As previously, the QBO, the PV and the ENSO variables indicate the role of large-scale dynamical processes whereas the $NO_2$ and HCHO variable indicate the role of chemistry and source reduction. Accounting for these variables, 90% of the negative trend observed with IASI between 2013 and 2016 is explained. The ENSO index with the large El Nino events in 2015-2016 as well as the decline of $NO_2$ tropospheric columns are the main contributions, 28% and 38% respectively, to explain the trend. Note that the residual trend is not significant and no additional trend is necessary. We did the test of introducing an additional trend but it does not reduce the residual trend and the p-value associated to the additional trend is very large (0.55). For the shorter time period 2013-2016, where the variations of the explicative variables are monotone, especially for $NO_2$ (Fig. 8), the linear regression model succeeds to explain most the observed trend. In this case, the regression model results suggest that dynamical processes as well as emission reduction contribute almost equally to the decreasing trend observed by IASI in the LT.

## 5 Conclusions

We use the IASI-A instrument to calculate the trends of LT ozone over the NCP during the nine-year period 2008-2016. However, questions remain on the reliability of tropospheric ozone trend derived from satellite observations. Indeed, a recent work comparing tropospheric ozone trends derived from IR and UV satellite sounders reveal inconsistencies (Gaudel et al., 2018), with IR sounders showing a general negative trend (Oetjen et al., 2015; Wespes et al., 2017a) and UV sounders a general positive trend (Cooper et al., 2014). The first step of our study was then to evaluate the stability and the reliability of the IASI ozone product used to calculate the trend. We explored the temporal evolution of the internal and external parameters, to which the retrieval is sensitive on the one hand, and on the other hand, we compare the IASI-A ozone

observations with independent measurements. As the thermal infrared observations are sensitive to the atmospheric thermal conditions, we evaluated the temporal evolution of the surface temperature and the thermal contrast over the NCP between 2008 and 2016. No specific and significant trend has been found. We also explored the influence of the changes in tropopause height on the LT ozone columns. No significant trend has been noticed in the tropopause height during the period. Coarse aerosol spectral features can contaminate the ozone spectral region used for the retrieval and then possibly affect the ozone retrieval. Filtering out observations associated with large aerosol loading (AOD > 0.2) does not change significantly the calculated trend from IASI observations. Thus, large aerosol loading that occurs regularly over China does not impact the trend derived from IASI. The stability of the retrieval has been evaluated using the averaging kernels and the associated parameters: the DOF and the altitude of maximum sensitivity. These two parameters do not show any significant trend. In addition, we performed a numerical experiment by considering a nine-year period with a constant ozone profile, and thus no trend. We applied the AK to the profile to evaluate the capability of the used IASI ozone product to reproduce this no-trend situation. No significant trend has been found in the resulting timeseries. Finally, we compared the LT ozone columns derived from IASI-A to independent observations. Comparison with the independent IASI-B observations over the 2013-2016 period shows similar trends. This indicates that no instrumental drift is responsible for the trend calculated from IASI-A observations. Comparisons with Asian ozonesondes show a bias ranging from -10 % to -15 %. The limited sampling and changes in the instrumentation of three sondes over five during the period do not allow one to evaluate clearly and firmly conclude concerning the reliability of IASI trends compared to those of the sondes. In a general way, comparisons with independent measurements (sondes or surface in situ) performed in this study show the importance of the sampling in the conclusions drawn. Differences in the sampling can affect significantly the calculated trends and thus the conclusions. One recommendation when comparing data sets with different sampling would be to perform the comparison over subsets of data having similar sampling.

According to the evaluation done, the trends derived from the IASI-A observations seem fairly reliable and can be used to study the LT ozone trend over the NCP. The analysis of the LT ozone columns shows a negative trend of ozone in the lower troposphere with 2013 being a pivotal year. Before 2013, no trend is detected whereas a significant negative trend of -0.24±0.06 DU/yr (-1.161±0.003 %/yr) is derived for 2013-2016. A similar trend is observed with the independent IASI-B instrument for the same period. Comparison with trends calculated for other partial columns (UT and TOC) shows that the trend derived for the LT is independent of other partial columns and well representative of the LT or more exactly of the free troposphere (3-5 km) where the used IASI ozone product is the most sensitive. We use a multivariate linear regression model to identify the processes driving the observed trend. The results suggest that both large-scale dynamical processes and regional emission changes explain the trend. At the end of the period (2013-2016), both contribute sensibly equally to the observed trends with the strong ENSO event in 2015-2016 and the $NO_x$ emission reduction being the largest contributors. For the entire period (2008-2016), the dynamical processes, especially a possible reduction of the STE, dominate to explain the nine-year trend. However, the entire trend is not explained by the linear regression model pointing out the difficulty to identify good proxies to characterize the role of advection and long-range transport and to account for non-linear processes

such as ozone chemistry. To properly evaluate these processes, use of chemistry-transport models is certainly needed but with the difficulty of having emission inventories including all the co-emitted species updated (especially VOCs species) over the entire hemisphere for a time period covering about 10 years. For example, using the CHIMERE model, we have been able to evaluate the response of ozone to the $NO_x$ emissions reduction, which is different depending on the altitude

(positive in the boundary layer and negative above 3 km). This explains, at least partly, the apparent inconsistency between the positive trend derived from the surface measurements and the negative trend derived in the lower/free troposphere from IASI. A better understanding and evaluation of the altitude-dependent ozone response to emission changes and the link with chemical regimes are still necessary. To do so, detailed modeling studies such as the one reported by Jin et al. (2017) but extended in altitude are necessary and require observations with a high vertical resolution such as those provided by aircraft

campaigns or the IAGOS program (Petzold et al., 2015).

**Data and code availability**

The IASI observations (level 1C) are available from the AERIS data infrastructure (www.aeris-data.fr). The IASI ozone product retrieved from the level 1C data is available, upon requested to Gaëlle Dufour (dufour@lisa.u-pec.fr), for the Asian domain considered here between 2008 and 2016.

The $NO_2$ and HCHO tropospheric columns from OMI are available at www.temis.nl.

The ozonesondes data are available from the WOUDC database (www.woudc.org), except for the Beijing station. These data can be requested directly to the PI of the station, Dr Y. Liu (liuyi@mail.iap.ac.cn). The surface measurements at the SDZ station are available upon request to the PIs of the station, Dr X. Xu (13311298991@189.cn) and W. Lin (linwl@muc.edu.cn).

The CHIMERE model is publicly available at www.lmd.polytechnique.fr/chimere.

**Author contribution**

GD managed the study from its conception, the formal analysis of data, the preparation of the manuscript and the funding acquisition. ME performed the IASI ozone retrieval and managed the resulting level-2 product. MB, JC, and GF participated to the conception of the study and in reviewing and editing the manuscript. AFC and ML performed the model simulations

and the emission corrections. WL and XX provided the surface observations, participated to the analysis and reviewed the manuscript. YL and YZ provided the ozonesonde measurements over Beijing, support for their analysis and reviewed the manuscript.

**Competing interests**

The authors declare that they have no conflict of interest.

**Acknowledgements**

The authors are grateful for the essential support of the Agence Nationale de la Recherche (ANR) through the PolEASIA
project (ANR-15-CE04-0005). The IASI mission is a joint mission of EUMETSAT and the Centre National d'Etudes
Spatiales (CNES, France). This study was financially supported by the French Space Agency – CNES (project
"IASI/TOSCA"). The authors acknowledge the AERIS data infrastructure (https://www.aeris-data.fr) for providing access to
the IASI Level 1C data, distributed in near-real-time by Eumetsat throught the EumetCast system distribution. The authors
acknowledge dataset producers and providers used in this study: the LATMOS/ULB for the provision of IASI CO total
columns through the AERIS database ; the ozonesonde data used in this study were mainly provided by the World Ozone
and Ultraviolet Data Centre (WOUDC) and are publicly available (see http://www.woudc.org) ; the NASA Giovani portal
for access to MODIS AOD products, the ERA Interim portal for access to meteorological fields needed to support our study.
We acknowledge the Institut für Meteorologie und Klimaforschung (IMK), Karlsruhe, Germany, for a licence to use the
KOPRA radiative transfer model. We also warmly thank O. R. Cooper from NOAA (US) for his support and fruitful
discussions as well as D. Hauglustaine from LSCE (France) for providing LMDZ-INCA model simulations.

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

**Table1: Statistics of the IASI and ozonesondes comparisons**

| Sonde[a] | 2008-2015 | | | | | 2008-2011 | | 2012-2015 | |
|---|---|---|---|---|---|---|---|---|---|
| | ndays | Bias (DU/%) | r | Slope[c] IASI | Slope[c] sonde | Bias (DU/%) | r | Bias (DU/%) | r |
| Naha | 186 | -3.2/-14 | 0.77 | -0.01±0.03 | 0.03±0.07 | -2.6/-11 | 0.84 | -3.8/-16 | 0.77 |
| Hong Kong | 188 | -2.5/-11 | 0.54 | 0.02±0.03 | 0.02±0.03 | -2.5/-11 | 0.69 | -2.5/-11 | 0.39 |
| Sapporo | 149 | -0.7/-3.2 | 0.07 | -0.006±0.04 | 0.056±0.04 | -0.28/-1.3 | 0.26 | -1.2/-4.9 | 0.11 |
| Tateno | 174 | -3.0/-12 | 0.81 | -0.002±0.04 | 0.07±0.05 | -2.4/-10 | 0.87 | -3.5/-14 | 0.75 |
| Beijing[b] | 106 | -7.9/-26 | 0.60 | -0.075±0.06 | -0.25±0.13 | | | | |
| Payerne | 257 | 0.53/2.8 | 0.17 | -0.014±0.03 | -0.04±0.02 | -0.01/-0.07 | 0.55 | 1.0/5.4 | 0.03 |
| w/o DJF | | -0.02/-0.12 | 0.68 | -0.03±0.02 | -0.03±0.02 | -0.18/-0.94 | 0.65 | 0.05/0.25 | 0.68 |

5  [a] the correction factor is not considered to filter the data (no significant changes), except for Beijing, where only sonde measurement with a correction factor ranged between 0.8 and 1.2 are considered.
[b] data are available only for the 2008-2014 period with a gap in 2013 due to instrumentation changes.
[c] the slope is calculated as the linear regression of the seasonal timeseries of IASI and smoothed sonde LT columns in Dobson unit.

**Table 2: Evolution of the residual trend and contribution of the explicative variables to the observed trend.**

| 2008-2016 | | | 2013-2016 | | |
|---|---|---|---|---|---|
| Variables included in the fit | Residual trend (DU/yr) | Contribution to the observed trend (%) | Variables included in the fit | Residual trend (DU/yr) | Contribution to the observed trend (%) |
| Observed trend | -0.17 | | Observed trend | -0.29 | |
| QBO | -0.15 | 12 | QBO | -0.26 | 10 |
| QBO+PV | -0.11 | 24 | QBO+ENSO | -0.18 | 28 |
| QBO+PV+ENSO | -0.08 | 18 | QBO+ENSO+PV | -0.15 | 10 |
| QBO+PV+ENSO+CO | -0.06 | 12 | QBO+ENSO+PV+NO2 | -0.04 | 38 |
| QBO+PV+ANSO+CO+trend | 0.005 | | QBO+ENSO+PV+NO2+HCHO | -0.03 | 3 |

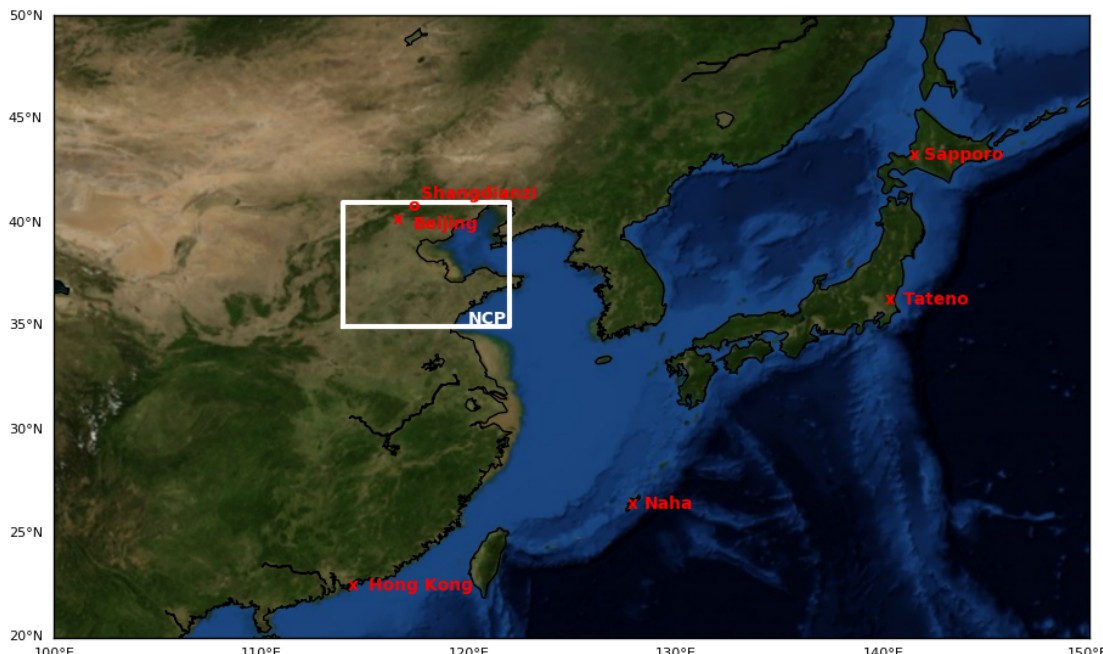

**Figure 1: Localization of the NCP region considered in the study indicated by the large square as well as the surface (o) and ozonesonde (x) stations.**

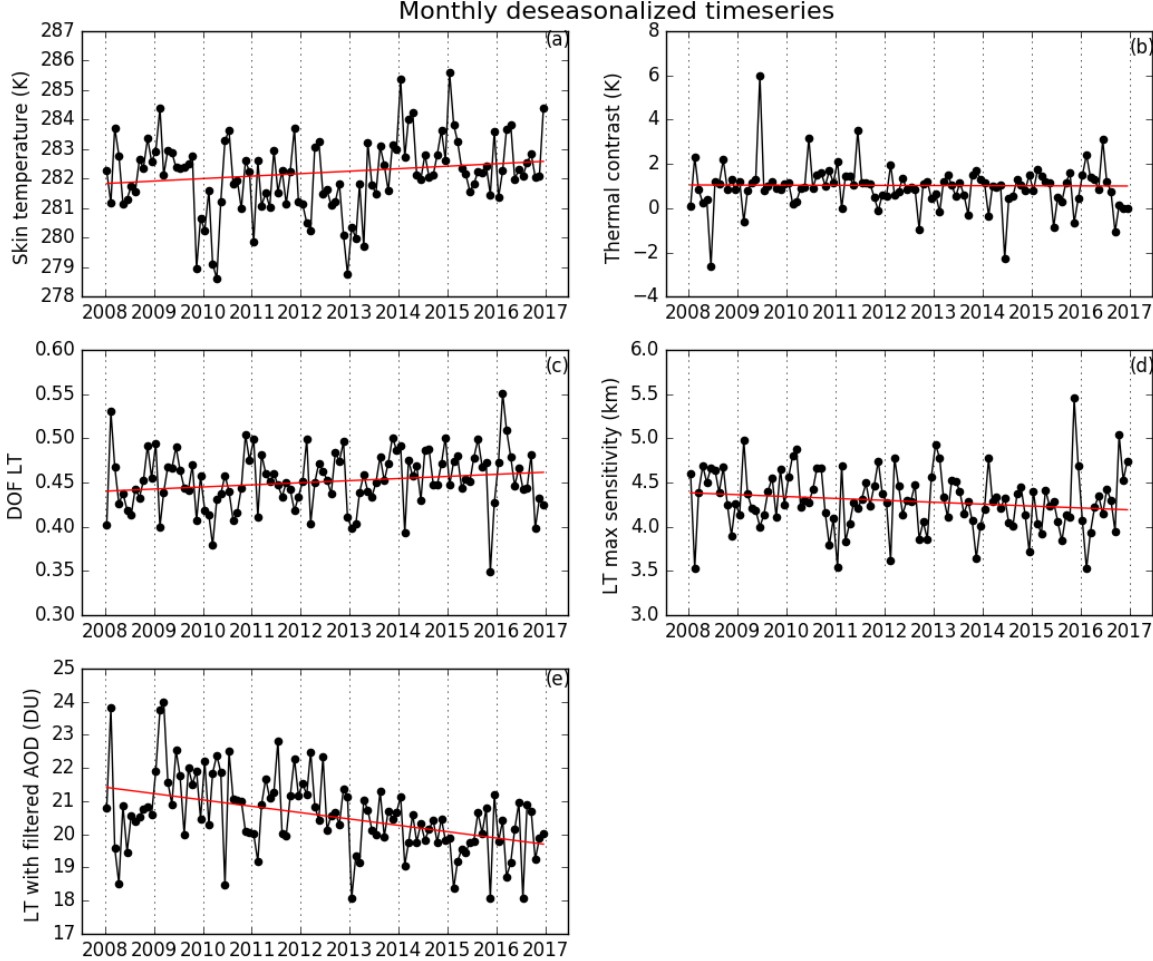

**Figure 2: Monthly deseasonalized timseries and their associated linear trends of internal and external parameters used to test the retrieval stability: (a) Skin temperature, (b) thermal contrast, (c) DOF, (d), altitude of maximum sensitivity, and (e) LT ozone filtered from large aerosol loading**

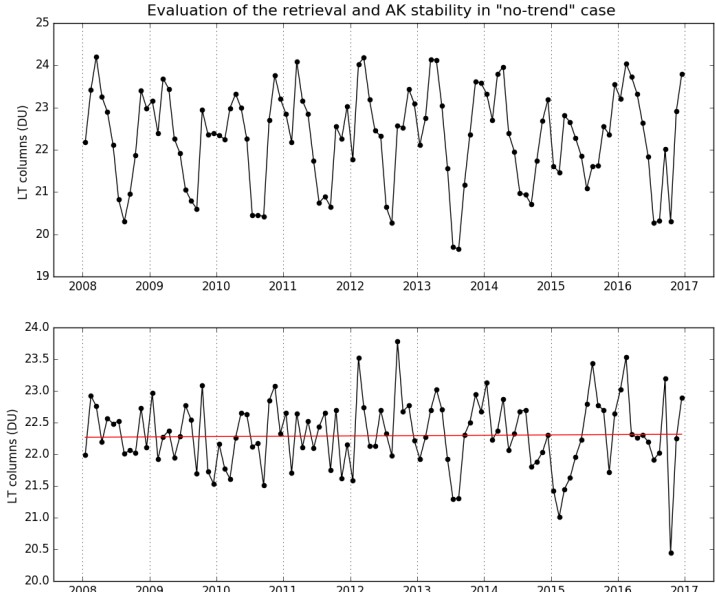

**Figure 3: Monthly timeseries (top) and its associated deseasonalized timeseries and linear trend calculated for a unique ozone profile smoothed by each individual averaging kernel of IASI over the NCP between 2008 and 2016.**

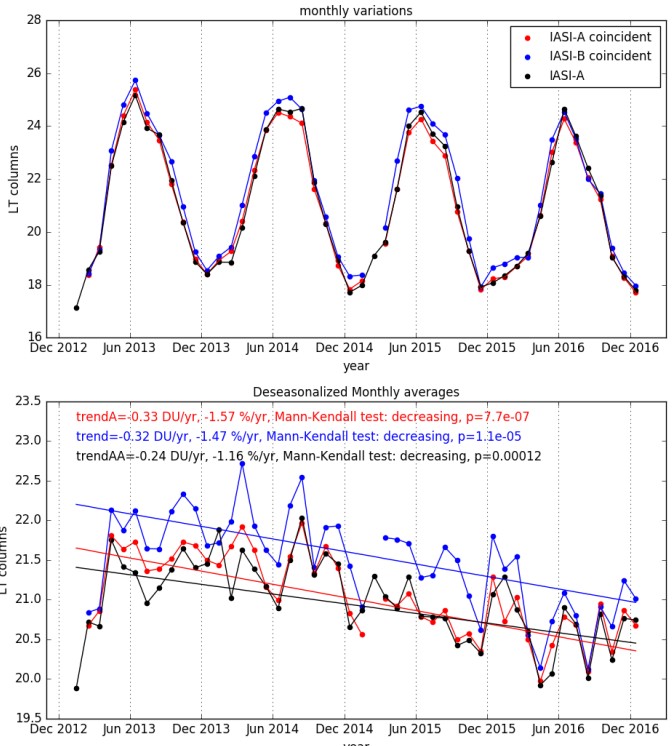

**Figure 4: Monthly timeseries of IASI-A and IASI-B (blue). The IASI-A timeseries is plotted in red when in coincidence with IASI-B and in black when all IASI-A observations are considered. The deseasonnalized timeseries and the corresponding linear trends are given in the bottom panel.**

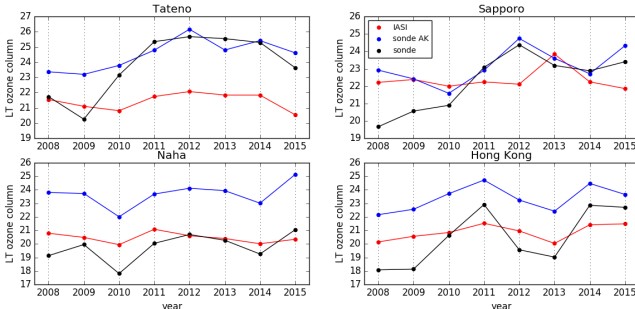

**Figure 5: Interannual variations of LT ozone columns observed by IASI (red) and measured by sondes with (blue) and without (black) applying the averaging kernels for four Asian stations (Tateno, Sapporo, Naha, and Hong Kong).**

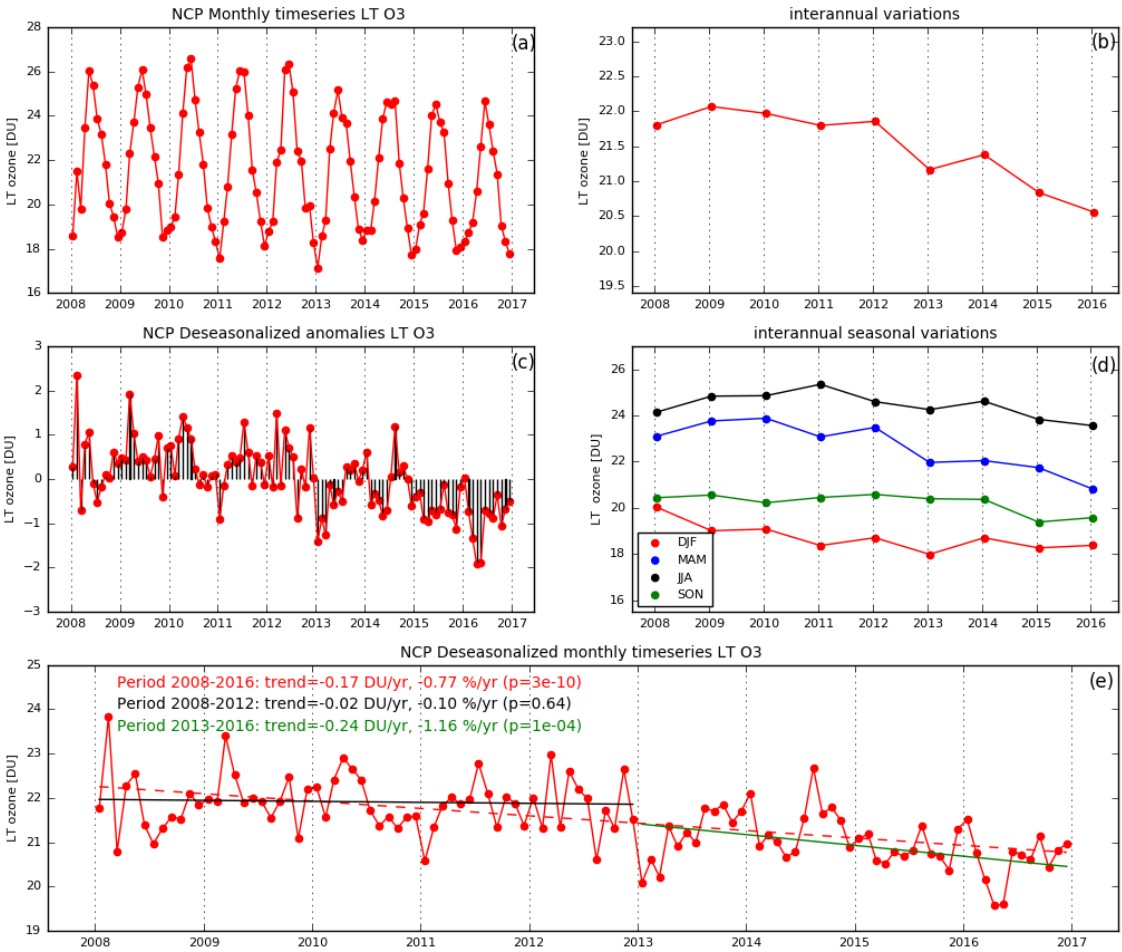

**Figure 6: Monthly, annual and seasonal evolution of LT ozone over the NCP between 2008 and 2016 derieved from IASI-A. (a) monthly timeseries, (b) annual timeseries, (c) anomalies, (d) interannual variation of seasonal means, (e) deseasonalized monthly timeseries with linear regression calculated for the 2008-2016 period (red), the 2008-2012 period (black), and the 2013-2016 period (green). The 2013 breakpoint of the deseasonalized timeseries (e) is chosen according to the significant change noticed in the annual timeseries (b) (see text for details).**

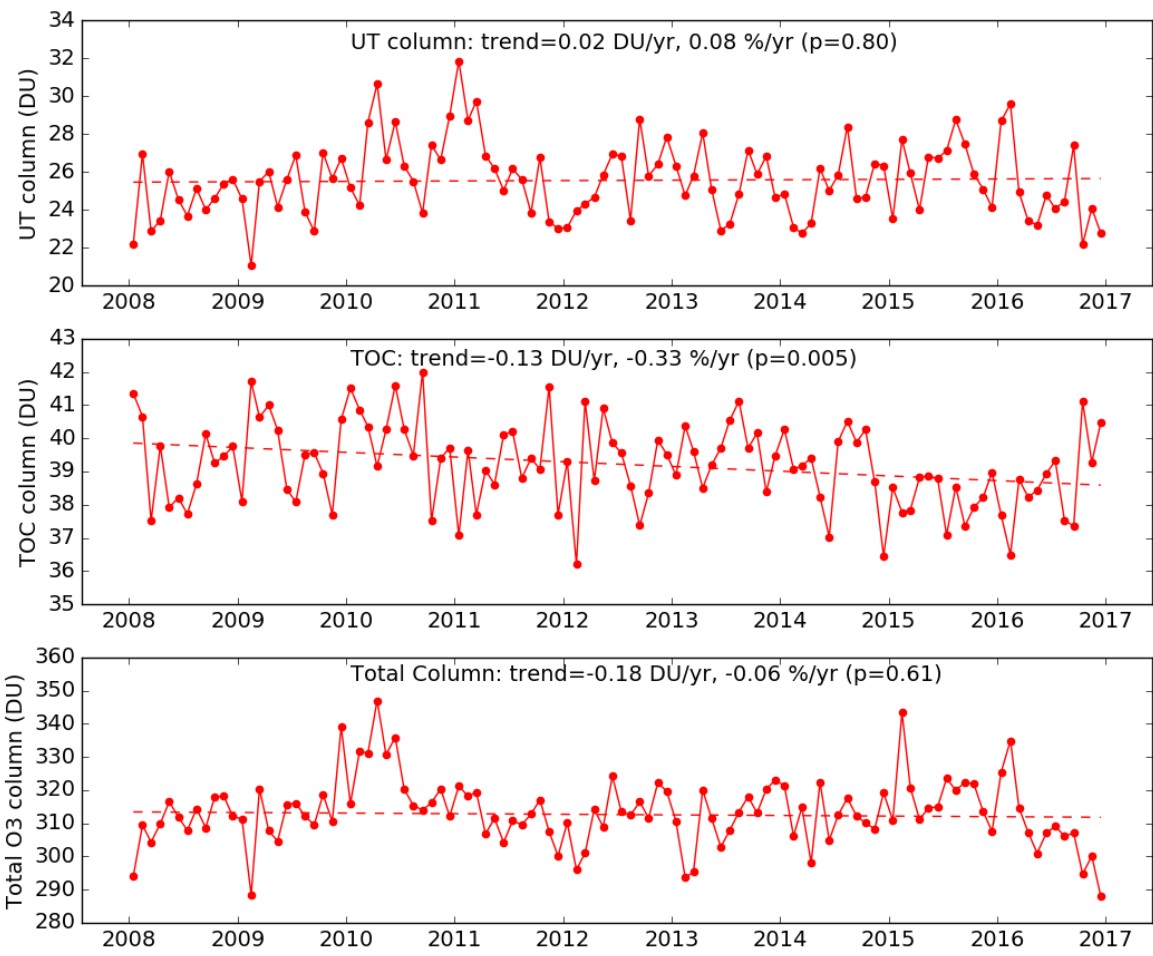

**Figure 7: Deseasonalized monthly timeseries of the upper tropospheric (UT) ozone column (top), the tropospheric ozone column (TOC, middle), and the total ozone column (bottom).**

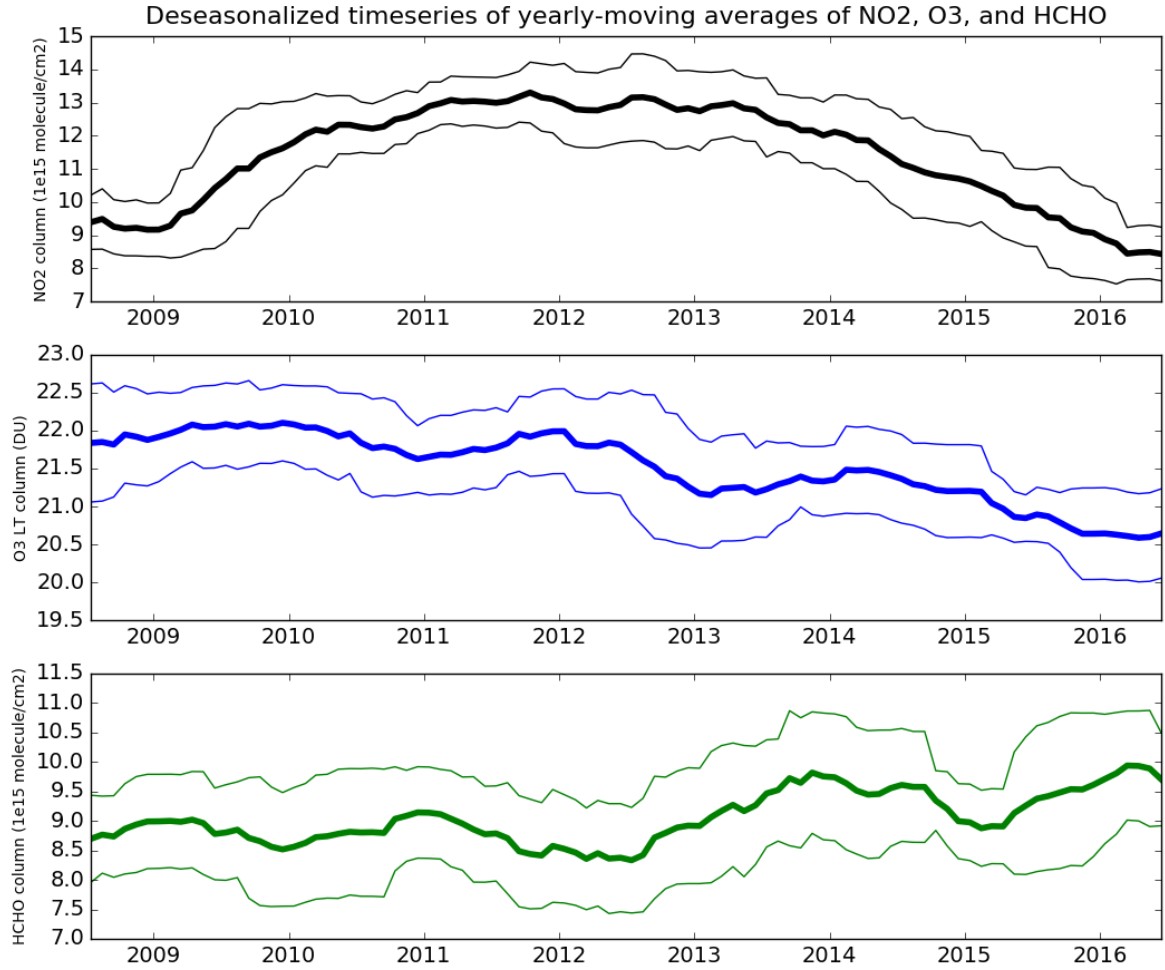

**Figure 8: Yearly-moving averages of the deseasonalized timeseries of OMI NO$_2$ tropospheric columns (top), IASI LT ozone columns (middle), and OMI HCHO tropospheric columns (bottom) over the NCP. The thinner lines represent the 1σ standard deviation range of the moving averages.**

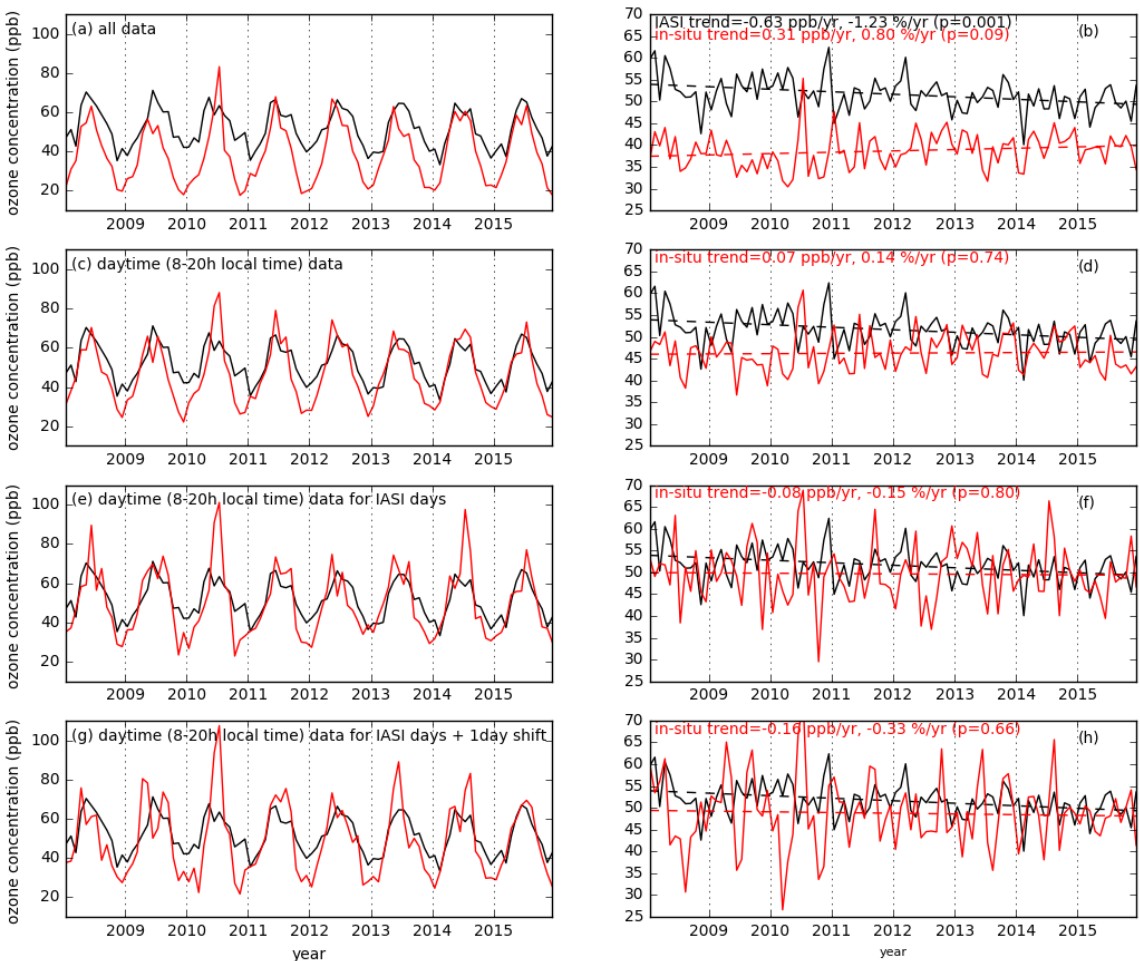

**Figure 9: Evolution of the timeseries (deseasonalized or not) when changing the sampling. The IASI equivalent column concentrations are plotted in black, the surface concentrations measured at the Shangdianzi station, China, in red.**

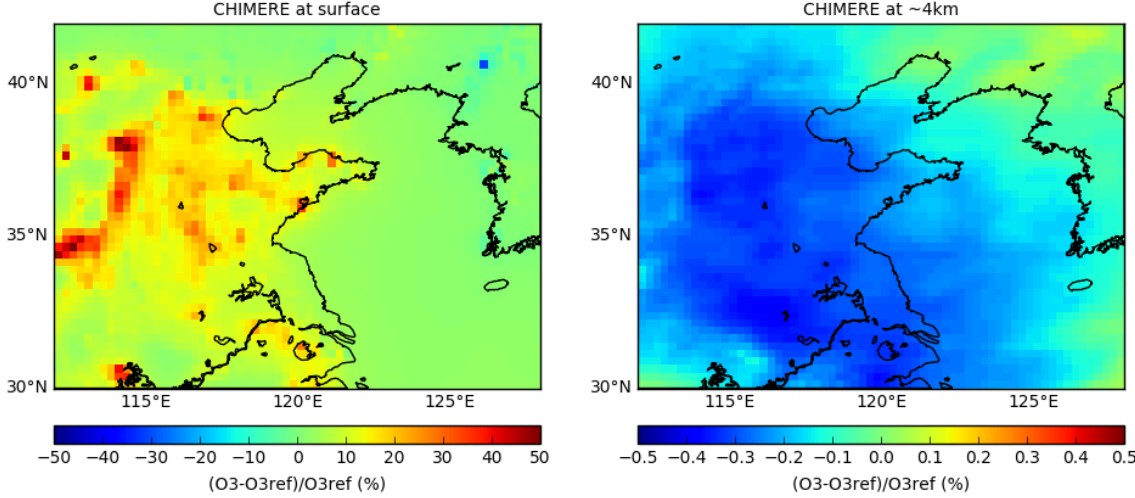

**Figure 10: Relative difference (%) at the surface and at ~4km between CHIMERE simulation based on corrected NO$_x$ and SO$_2$ emissions using OMI satellite data and CHIMERE simulation based on EDGAR-HTAP-v2.2 2010 emission inventory.**

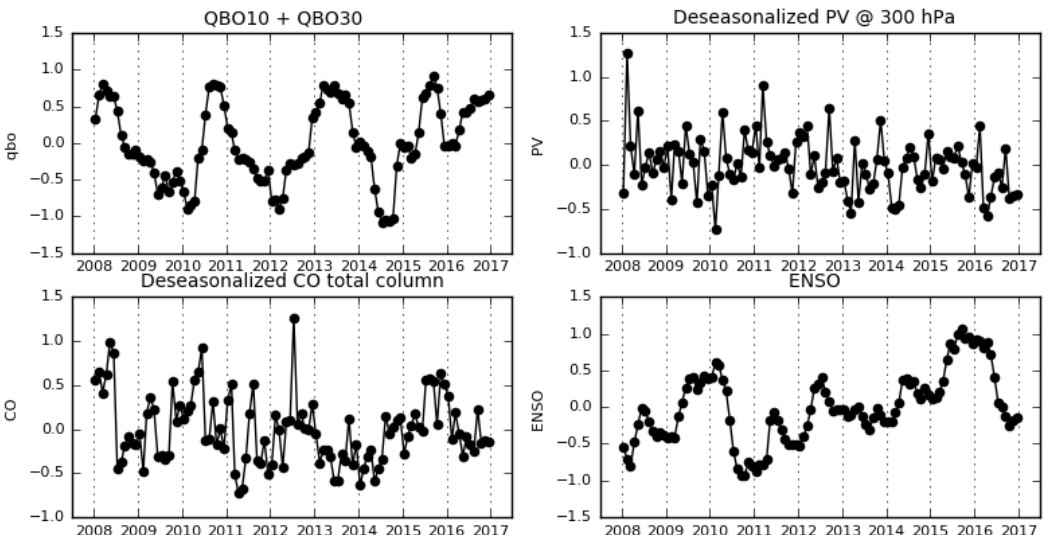

**Figure 11: Normalized explanatory variable timeseries between 2008 and 2016.**

