# Peer review of "Lower tropospheric ozone over the North China Plain: variability and trends revealed by IASI satellite observations for 2008-2016"

_Atmospheric Chemistry and Physics, 2018_

## Referee Comment (RC1) · Anonymous Referee #1 · 2 Jul 2018

* Overall an interesting and relevant paper. The data are well presented, the measuring and analysis methods seems sound, but the text could greatly benefit from an extra text editing. I also suggest some corrections below.

* The negative trend of ozone in the lower troposphere from IASI compared to other satellite based products makes me frown a bit. The authors try to provide a reasonable explanation for this, but to my opinion, it remains inconclusive. It makes me uneasy in assessing this manuscript, although the analysis made on the data seems to be sound. The authors analyzed different possible factors but in the end a remaining -0.1 DU/yr trend remains unexplained, which is more than half the trend (-0.17 DU/yr).

[Figure]

The only argument for understanding the negative trend is based on a speculation that increasing VOC emissions at the surface are responsible for the increasing trend of ozone observations near the surface from ground stations. But the authors do not show data that would demonstrate that. What is more, this VOC's hypothesis does not explain why the UV based sensors do observe a positive trend. To my knowledge these sensors are also not very sensitive for ozone near the surface. This is really puzzling. I would suggest to elaborate a bit on this assumption by looking to for instance formaldehyde data (see paper Jin et al., 2017, JGR: 10.1002/2017JD026720).

* What happens if the whole tropospheric column of IASI is considered, instead of LT? Which trends are then revealed? And what if one only considers the upper troposphere, UL? Which trends are then computed? Could this add some more support in understanding the negative trend?

* Is there any trend in the thermal contrast retrieved from the IASI data?

* In the deseasonalized data there seems to be a dip in LT ozone at the start (January?) of 2011 (fig 2(e), fig 4. Is there any explanation for this?

P1L14-15: "... decrease in NO2 tropospheric columns since 2013 attributed to ...";

P1L16: "... remains unclear.";

P1L19: "... leading to an overall significant trend of ...";

P1L21: "... from IASI may be attributed to a reduction ...";

P1L23: "... applied CO proxy.";

P1L23: "... from background surface ozone(?) measurements ...";

P1L26: "..., without any conclusive explanation so far.";

P1L28-29: "... from the comparison concerns the impact of the spatial and temporal sampling of the datasets on the calculated trends.";

P1L31: "... increasing pollutant ... ";

P2L2: remove "...for the major pollutants"

P2L3: "... attributed to the emissions increase to both surface as well as in the lower troposphere...";

P2L25: "... as processes that modify...";

P3L3-5: Rephrase this sentence. Needs at least a proper verb;

P3L8: use " with respect to" instead of "in regards";

P3L11 and 18, etc: use "operational" instead of "in flight"; also check and replace in other parts;

P4L29-30: This method is based on the difference between the actual month and the average value for that month for the period 2008-2016?

P4L32: So when I see a computed linear trend in the text, figures or tables it is always based on the Theil-Sen estimator? If not please specify.

P5L32: use "etc" instead of "...";

P6L2: " ... one region, ...";

P6L3: "...have been observed in recent years. Thus, the hypotheses is that reductions in surface emissions of NOX might cause a decreasing trend in lower tropospheric ozone levels.";

P6L7: introduce white spaces before and after the equation;

P6L10-11: "The significance of including or excluding a variable is ...";

P6L13: "Variable that were not significant were remove from the final fit.";

P6L18: "... from daily data, ...";

P6L30: I believe that it is "... have also been tested. ";

P6L32: "were";

P6L34: "After the fitting procedure, the significant variables are: ...";

P6L35: "The normalized ...";

P7L30: "... changes in emissions addressed by CO as proxy.";

P7L30-31: I do not understand what the authors try to say. Please clarify and rephrase. Transport effects?

P7L34: "more" should be at the end of the sentence;

P8L6: "... but requires up-to-date emissions inventories ... "; this can be achieved by using OMI NO2 and other data;

P8L19: "However, time series of monthly skin temperature show ...";

P8L29: "... calculated insignificant trend of ...";

The linear trends computed in Figure 5: are they based on the deseasonalized data? If not, why not? They should! Please clarify.

P9L13: "... but not significant"; since p> 0.05;

P9L14-16: This is a bad sentence. Please rephrase!

P10L4-5: "... might not be attributed to... ;"

P10L15: do you mean "relaxed" instead of released ?

P10L33: "small bias";

P11L23: "... can completely change...";

P12L1-2: "... (daily, hourly)...";

[Figure]

Figure 2: Please add on the y-axis "Deseasonalized" LT ozone; Add in caption how the 2013 breakpoint is chosen;

Figure 5: Are the associated linear trends based on the monthly timeseries or on the deseasonalized series?

Figure 5. Please add the deseasonalized timeseries of the thermal contrast!

Figure 8. Should be "four stations";

———————————————————

---

## Referee Comment (RC2) · Anonymous Referee #2 · 15 Jul 2018

This manuscript deals with an important question of current interest, i.e., the discrepancy of O3 trends over China derived from different satellite sensors (UV IR), and surface measurements. Specifically, trends derived from the IASI instruments are negative, while those derived from ground stations are positive. The authors address the following issues: a) attribution of trends to both changes in emissions and meterological parameters; b) examination of the stability and robustness of the trends derived from IASI and implications for derived trends. Overall, this is a strong manuscript. The authors have been very thorough in addressing the different issues involved, and their approach sets up a path for future data analyses and modeling efforts.

[Figure]

A general comment is that, given the multiplicity of issues discussed, the Conclusions could be more detailed in terms of the specific questions addressed. These conclusions are dispersed in the different sections, but it would help if a summary is given as to, for example, conclusion on attributions, impact of retrieval parameters, comparison to sondes and surface stations, and issues of sampling. Most important, after this study, what is the status of the original question posed above?

Specific comments:

1. Abstract, line 24: Replace "as well" with "similarly"

2. Abstract: It is very descriptive of the different issues/approaches, but it seems to be missing an obvious conclusion (even if it is a "negative" one).

Page 2, line 7: Somewhere in the manuscript there should be more said about the potential discrepancy with MOZAIC data, particularly since the hypothesis is proposed that different regions of the troposphere are NOX or VOC limited. I presume that a lot of the MOZAIC data comes from higher altitudes in the LT, which would also be where IASI retrievals are most sensitive.

Page 4, Section 3.1: Discussion of the time series analysis should have some clarifications. References should be given for "Theil-Sen estimator" and "Mann-Kendall test." Use of certain terms, such as "climatological index" are unclear, and should be quantitatively defined through an equation, and not words.

Page 5, line 6: "amplitude", not "altitude"

Page 5, lines 11-12, and elsewhere: The authors note the change in O3 in 2013, but it is not clear why this happens. If we do not have a reason it should be stated as an outstanding question.

Section 3.3, explicative variables: This is a very good discussion. One item that should be discussed in more detail is why the NOx abundances in the LT are not correlated with the O3. I would expect some correlation away from the surface?

Page 7, line 10: "A period for which strong stratospheric intrusions..."

Figure 4 and Table 1: I found Figure 4 difficult to interpret. The residuals are, of course, all over the place, and impacts are sometimes difficult to see, for example, the impact of PV on residuals at beginning of 2003 (it's only one point!). The general trends become slowly apparent, but the large variations in residuals mask some of the impact of the different explicative variables. I find that Table 1 does a much better job of succinctly summarizing the results, which is harder to get from Figure 4. One suggestion for Table 1, etc., is that the same analysis be carried out for the attribution to the trends in the 2013-2016 period. Given that this period has a large impact on the overall trend, it would be interesting to see how the attributions work for this period.

Page 9, lines 5-6. "similar to the trend derived from the data with no filters'?

Page 9, lines 22-25 and Figure 6. What is being plotted in figure 6, i.e., what are the units for the "AK stability" in the y-axis?

Page 10, line 4: Should read: THUS, the comparison of IASI-A and B...

Page 10, line 15: "The criterion on the time difference has been RELAXED

Page 12, line 8: "We consider daytime surface observations only ON the days

Page 12, line 10: "a recent study shows THAT the downward mixing...

---

## Author Comment (AC1) · 7 Sep 2018

Lower tropospheric ozone over the North China Plain: variability and trends revealed by IASI satellite observations for 2008-2016. Dufour et al.

The authors thank the referees for their interest in the article. Their suggestions, recommendations and remarks were very useful for improving the study and clarifying the conclusions. Additional work has been done and leads to major changes in the manuscript. The new structure of the manuscript is described after the response to the referee's comments. In the following referees' comments are indicated in italics and the reply to each comment is given just below. Quoted text from the revised manuscript is given in blue.

**Response to Referee #1 comments**

*\* The negative trend of ozone in the lower troposphere from IASI compared to other satellite based products makes me frown a bit. The authors try to provide a reasonable explanation for this, but to my opinion, it remains inconclusive. It makes me uneasy in assessing this manuscript, although the analysis made on the data seems to be sound. The authors analyzed different possible factors but in the end a remaining - 0.1 DU/yr trend remains unexplained, which is more than half the trend (-0.17 DU/yr). The only argument for understanding the negative trend is based on a speculation that increasing VOC emissions at the surface are responsible for the increasing trend of ozone observations near the surface from ground stations. But the authors do not show data that would demonstrate that. What is more, this VOC's hypothesis does not explain why the UV based sensors do observe a positive trend. To my knowledge these sensors are also not very sensitive for ozone near the surface. This is really puzzling. I would suggest to elaborate a bit on this assumption by looking to for instance formaldehyde data (see paper Jin et al., 2017, JGR: 10.1002/2017JD026720).*

We elaborate more on the assumption concerning the VOCs and the possible role of chemical regime changes with altitude. Dedicated analyses of OMI HCHO timeseries as well as OMI $NO_2$ timeseries shows that while $NO_2$ decreases since 2013, HCHO increases (Fig. 8 of the revised manuscript, reproduced below).

[Figure]

**Figure 8: Yearly-rolling averages of the deseasonalized timeseries of OMI $NO_2$ tropospheric columns (top), IASI LT ozone columns (middle), and OMI HCHO tropospheric columns (bottom).**

A very recent study (Zheng et al., https://doi.org/10.5194/acp-2018-374) about updated Chinese emission inventories shows that anthropogenic emissions of NMVOCs increase between 2010 and 2014 and stagnate since 2014. In addition, we used model simulations performed by the CHIMERE CTM to evaluate the ozone sensitivity to $NO_x$ emission changes (VOCs emissions kept constant). The discussion of the

results done in the revised manuscript is: "To better understand the changes of ozone at different altitudes over the NCP, we use simulation experiments of the chemistry-transport model CHIMERE (Menut et al., 2013) made in the framework of another study (Lachâtre et al., in prep[1]). Two runs of CHIMERE with different emissions are compared for the year 2015. The first one was performed based on the EDGAR-HTAP-v2.2 2010 emission inventory (Janssens-Maenhout et al., 2015) and considered as the reference case. For the second run, the $SO_2$ and $NO_2$ OMI tropospheric columns were used to update the $SO_2$ and $NO_x$ emissions using a simple mass-balance method for the emission correction. The corrected emissions include then the reduction of $NO_x$ emissions occurring the last years. Note that the VOCs emissions are the same for the two simulations. Figure 10 shows the differences (%) between the annual mean ozone concentration simulated with updated emissions and with the reference case at the surface level and at ~4km. At the surface, the ozone concentrations simulated with reduced $NO_x$ emissions are larger by 13% on average over the NCP. This corroborates the reported ozone increase, associated to the $NO_x$ emission reduction. On the contrary, the ozone concentrations at 4 km decrease compared to the reference when $NO_x$ emissions are reduced. The impact is small, -0.25 % on average over the NCP but it persists in the altitude range between 3 and 7 km, the range where the IASI observations are the most sensitive. These results suggest that our hypothesis concerning the response of LT or free tropospheric ozone (decrease) to the $NO_x$ reduction is credible and likely associated to the chemical regime changing from VOC-limited in the boundary layer to $NO_x$-limited in the free troposphere. Quantifying the change of chemical regime with the altitude is out of the scope of this study and would require observations with a better vertical resolution than those offered by satellite observations such as those from the IAGOS program (Petzold et al., 2015) and detailed model studies. The changes due to the $NO_x$ emission reduction on free tropospheric ozone remain small (Fig. 10) and do not allow to explain by themselves the negative trend observed with IASI. In the next section, we explore the processes contributing to the ozone variability and trend in addition to the $NO_x$ emission reduction."

[Figure]

**Figure 10: Relative difference (%) at the surface and at ~4km between CHIMERE simulation based on corrected $NO_x$ and $SO_2$ emissions using OMI satellite data and CHIMERE simulation based on EDGAR-HTAP-v2.2 2010 emission inventory.**

Following the recommendations of Referee #2, we applied the multivariate linear regression model not only for the entire period but also for the 2013-2016 period. Interestingly, the regression model allows one to explain 65% of the trend for the entire period but for 2013-2016 "90% of the negative trend observed with IASI between 2013 and 2016 is explained. The ENSO index with the large El Nino events in 2015-2016 as well as the decline of $NO_2$ tropospheric columns are the main contributions, 28% and 38% respectively, to explain the trend". Note that during 2008-2012, the IASI observations do not show any
* * *
[1] Lachatre, M., Fortems-Cheiney, A., Beekmann, M., Foret, G., Dufour, G., and Siour, G: The unintended consequence of $SO_2$ and $NO_2$ regulations over China: increase of ammonia levels and impact on $PM_{2.5}$ concentrations, in prep.

trend. The 2008-2016 negative trend is then driven by the 2013-2016 negative trend. Note that the $NO_2$ variable becomes not significant for 2008-2016 in the regression analysis. Indeed the $NO_2$ variations are not monotone (maximum in 2011/2012) over the entire period whereas the ozone variations are "more monotone" and decreasing. This points out the difficulties of the linear regression models to account for non-linear processes such as ozone chemistry and stress the need to use chemistry-transport models to evaluate the processes driving the ozone trend.

Concerning the referee's comment about the UV sensors. We agree that the discrepancy between UV and IR sounders is puzzling considering none are very sensitive to the surface. The TOAR community will likely work in the near future to address this question, which is out of the scope of this paper. In this study, we make a substantial effort to evaluate the stability and reliability of the IASI ozone product used to derive reliable trends.

*\* What happens if the whole tropospheric column of IASI is considered, instead of LT? Which trends are then revealed? And what if one only considers the upper tropo-sphere, UL? Which trends are then computed? Could this add some more support in understanding the negative trend?*
In the revised manuscript, we added information concerning timeseries and trends of the UT column, the tropospheric column and the total column as follows: "In order to evaluate if the LT columns can be strongly contaminated by the altitudes higher in the troposphere and the stratosphere, we also derive the trends for different partial columns: one ranging from 6 to 12 km and considered as the upper tropospheric column (UT), the tropospheric ozone column (TOC) ranging from the surface up to the tropopause, and the total column. Note that the UT column can include part of the lower stratosphere when the tropopause is lower than 12 km. The deseasonalized timeseries are plotted in Fig. 7 with the derived trends indicated in the figure. The UT and total columns do not show any trend whereas the TOC column presents a significant negative trend, likely driven by the negative trend observed in the lower troposphere. These results show that the negative trend observed in the lower troposphere with IASI is likely representative of the ozone evolution within the lower and more likely the free troposphere (3-5 km) where the IASI retrieval is the most sensitive." This allows one to conclude that the negative trend observed in the LT column is representative of this region of the atmosphere and not of altitudes above.

*\* Is there any trend in the thermal contrast retrieved from the IASI data?*
No trend is calculated from the thermal contrast retrieved from IASI. The corresponding deseasonalized timeseries is now plotted in Fig. 2 (former Fig. 5).

*\* In the deseasonalized data there seems to be a dip in LT ozone at the start (January?) of 2011 (fig 2(e), fig 4. Is there any explanation for this?*
It might be explained by the sampling of IASI that is less important for this month compared to the surrounding months.

*P1L14-15: ". . . decrease in NO2 tropospheric columns since 2013 attributed to . . ."; done*

*P1L16: ". . . remains unclear.";* done

*P1L19: ". . . leading to an overall significant trend of . . .";* done

*P1L21: ". . . from IASI may be attributed to a reduction . . .";* done

*P1L23: ". . . applied CO proxy.";* done

*P1L23: ". . . from background surface ozone(?) measurements ...";* done

*P1L26: ". . ., without any conclusive explanation so far.";* done

*P1L28-29: ". . . from the comparison concerns the impact of the spatial and temporal sampling of the datasets on the calculated trends.";* done

*P1L31: ". . . increasing pollutant . . . ";* done

*P2L2: remove ". . .for the major pollutants"* done

*P2L3: ". . . attributed to the emissions increase to both surface as well as in the lower troposphere. . .";* done

*P2L25: ". . . as processes that modify. . .";* done

*P3L3-5: Rephrase this sentence. Needs at least a proper verb;* done

"Trends derived from IR sounders are mainly negative, whereas they are positive when derived from UV sounders (Gaudel et al., 2018). One hypothesis to explain this discrepancy relies on the difference in vertical sensitivity."

*P3L8: use " with respect to" instead of "in regards";* done

*P3L11 and 18, etc: use "operational" instead of "in flight"; also check and replace in other parts;*

*P4L29-30: This method is based on the difference between the actual month and the average value for that month for the period 2008-2016?*
The method used is better described in the revised manuscript:

"We also calculate the deseasonalized monthly timeseries using the average-percentage method. A climatological index, $sindex$, is calculated over the considered nine-year period following Eq. (1).

$$sindex(im) = \frac{1}{9}\sum_{iy=1}^{9}\frac{month\_ave(im,iy)}{year\_ave(iy)} \qquad (1)$$

where $month\_ave$ is the monthly average for the NCP calculated as described previously for each month ($im$) and each year ($iy$), and $year\_ave$ is the yearly average. The climatological index is then applied to the monthly timeseries to remove the seasonal component from the series and obtained the deseasonalized timeseries $deseas$ (Eq. (2)).

$$deseas(im,iy) = \frac{month\_ave(im,iy)}{sindex(im)} \qquad (2)"$$

*P4L32: So when I see a computed linear trend in the text, figures or tables it is always based on the Theil-Sen estimator? If not please specify.*
Yes, it is. We added this sentence in the method description: "All the linear trends presented in the current study are computed based on the deseasonalized timeseries."

*P5L32: use "etc" instead of ". . .";* done

*P6L2: " . . . one region, . . .";* done

*P6L3: ". . .have been observed in recent years. Thus, the hypotheses is that reductions in surface emissions of NOX might cause a decreasing trend in lower tropospheric ozone levels.";* done

*P6L7: introduce white spaces before and after the equation;* done

*P6L10-11: "The significance of including or excluding a variable is . . .";* done

*P6L13: "Variable that were not significant were remove from the final fit.";* done

*P6L18: ". . . from daily data, . . ."; done*

*P6L30: I believe that it is ". . . have also been tested. ";* done

*P6L32: "were";* done

*P6L34: "After the fitting procedure, the significant variables are: . . .";* done

*P6L35: "The normalized . . .";* done

*P7L30: ". . . changes in emissions addressed by CO as proxy.";* done

*P7L30-31: I do not understand what the authors try to say. Please clarify and rephrase. Transport effects?*

"The CO variable, due to its long lifetime, is considered as a proxy for large-scale emission changes that may affect LT ozone regionally. To account for the long-range transport and advection, the regression model has been tested either with the CO timeseries averaged for the Northern Hemisphere or with the CO timeseries averaged only over the NCP."

*P7L34: "more" should be at the end of the sentence;* done

*P8L6: ". . . but requires up-to-date emissions inventories . . . "; this can be achieved by using OMI NO2 and other data;*
Yes, we agree but up to now no updated emissions inventories (with consistent co-emitted species) are available at the global scale over the almost 10-year period considered here. That is why we stated this difficulty. In the revised version of the manuscript, we precise this as follows: "use of chemistry-transport models is certainly needed but with the difficulty of having emission inventories including all the co-emitted species updated (especially VOCs species) over the entire hemisphere for a time period covering about 10 years"

*P8L19: "However, time series of monthly skin temperature show . . .";* done

*P8L29: ". . . calculated insignificant trend of . . .";* done

*The linear trends computed in Figure 5: are they based on the deseasonalized data? If not, why not? They should! Please clarify.*
Yes, they are. To avoid confusion, we modified the figure and now present the deseasonalized timeseries (including the thermal contrast retrieved from IASI).

*P9L13: ". . . but not significant"; since p> 0.05;* done

*P9L14-16: This is a bad sentence. Please rephrase!*
In the revised version of the manuscript, this sentence was suppressed.

*P10L4-5: ". . . might not be attributed to. . . ;"* done

*P10L15: do you mean "relaxed" instead of released ?* changed

*P10L33: "small bias";* done

*P11L23: ". . . can completely change. . .";* done

*P12L1-2: ". . . (daily, hourly). . .";* done

*Figure 2: Please add on the y-axis "Deseasonalized" LT ozone; Add in caption how the 2013 breakpoint is chosen;*
The "deseasonalied" LT ozone is indicated in the subtitle of the figure. Concerning the 2013 breakpoint: "The 2013 breakpoint of the deseasonalized timeseries (e) is chosen according to the significant change noticed in the annual timeseries (b) (see text for details)"

*Figure 5: Are the associated linear trends based on the monthly timeseries or on the deseasonalized series?*
See above

*Figure 5. Please add the deseasonalized timeseries of the thermal contrast!*
See above

*Figure 8. Should be "four stations";* done

**Response to Referee #2 comments**
*A general comment is that, given the multiplicity of issues discussed, the Conclusions could be more detailed in terms of the specific questions addressed. These conclu-sions are dispersed in the different sections, but it would help if a summary is given as to, for example, conclusion on attributions, impact of retrieval parameters, compari-son to sondes and surface stations, and issues of sampling. Most important, after this study, what is the status of the original question posed above?*
The conclusions have been rewriten and read now as follows: " We use the IASI-A instrument to calculate the trends of LT ozone over the NCP during the nine-year period 2008-2016. However, questions remain on the reliability of tropospheric ozone trend derived from satellite observations. Indeed, a recent work comparing tropospheric ozone trends derived from IR and UV satellite sounders reveal inconsistencies (Gaudel et al., 2018), with IR sounders showing a general negative trend (Oetjen et al., 2015; Wespes et al., 2017a) and UV sounders a general positive trend (Cooper et al., 2014). The first step of our study was then to evaluate the stability and the reliability of the IASI ozone product used to calculate the trend. We explored the temporal evolution of the internal and external parameters, to which the retrieval is sensitive on the one hand, and on the other hand, we compare the IASI-A ozone observations with independent measurements. As the thermal infrared observations are sensitive to the atmospheric thermal conditions, we evaluated the temporal evolution of the surface temperature and the thermal contrast over the NCP between 2008 and 2016. No specific and significant trend has been found. We also explored the influence of the changes in tropopause height on the LT ozone columns. No significant trend has been noticed in the tropopause height during the period. Coarse aerosol spectral features can contaminate the ozone spectral region used for the retrieval and then possibly affect the ozone retrieval. Filtering out observations associated with large aerosol loading (AOD > 0.2) does not change significantly the calculated trend from IASI observations. Thus, large aerosol loading that occurs regularly over China does not impact the trend derived from IASI. The stability of the retrieval has been evaluated using the averaging kernels and the associated parameters: the DOF and the altitude of maximum sensitivity. These two parameters do not show any significant trend. In addition, we performed a numerical experiment by considering a nine-year period with a constant ozone profile, and thus no trend. We applied the AK to the profile to evaluate the capability of the used IASI ozone product to reproduce this no-trend situation. No significant trend has been found in the resulting timeseries. Finally, we compare the LT ozone columns derived from IASI-A to independent observations. Comparison with the independent IASI-B observations over the 2013-2016 period shows similar trends. This indicates that no instrumental drift is responsible for the trend calculated from IASI-A observations. Comparisons with Asian ozonesondes show a bias ranging from -10 % to -15 %. The limited sampling and changes in the instrumentation of three sondes over five during the period do not allow one to evaluate clearly and firmly conclude concerning the reliability of IASI trends compared to those of the sondes. In a general way, comparisons with independent measurements (sondes or surface in situ) performed in this study show the importance of the sampling in the conclusions drawn. Differences in the sampling can affect significantly the calculated trends and thus the conclusions. One recommendation when comparing data sets with different sampling would be to perform the comparison over subsets of data having similar sampling.
According to the evaluation done, the trends derived from the IASI-A observations seem fairly reliable and can be used to study the LT ozone trend over the NCP. The analysis of the LT ozone columns shows a

negative trend of ozone in the lower troposphere with 2013 being a pivotal year. Before 2013, no trend is detected whereas a significant negative trend of -0.24±0.06 DU/yr (-1.161±0.003 %/yr) is derived for 2013-2016. A similar trend is observed with the independent IASI-B instrument for the same period. Comparison with trends calculated for other partial columns (UT and TOC) shows that the trend derived for the LT is independent of other partial columns and well representative of the LT or more exactly of the free troposphere (3-5 km) where the used IASI ozone product is the most sensitive. We use a multivariate linear regression model to identify the processes driving the observed trend. The results suggest that both large-scale dynamical processes and regional emission changes explain the trend. At the end of the period (2013-2016), both contribute sensibly equally to the observed trends with the strong ENSO event in 2015-2016 and the $NO_x$ emission reduction being the largest contributors. For the entire period (2008-2016), the dynamical processes, especially a possible reduction of the STE, dominate to explain the nine-year trend. However, the entire trend is not explained by the linear regression model pointing out the difficulty to identify good proxies to characterize the role of advection and long-range transport and to account for non-linear processes such as ozone chemistry. To properly evaluate these processes, use of chemistry-transport models is certainly needed but with the difficulty of having emission inventories including all the co-emitted species updated (especially VOCs species) over the entire hemisphere for a time period covering about 10 years. For example, using the CHIMERE model, we have been able to evaluate the response of ozone to the $NO_x$ emissions reduction, which is different depending on the altitude (positive in the boundary layer and negative above 3 km). This explains, at least partly, the apparent inconsistency between the positive trend derived from the surface measurements and the negative trend derived in the lower/free troposphere from IASI. A better understanding and evaluation of the altitude-dependent ozone response to emission changes and the link with chemical regimes are still necessary. To do so, detailed modeling studies such as the one reported by Jin et al. (2017) but extended in altitude are necessary and require observations with a high vertical resolution such as those provided by aircraft campaigns or the IAGOS program (Petzold et al., 2015)."

*Specific comments:*

*\* Abstract, line 24: Replace "as well" with "similarly"*
Due to revisions, this part has been suppressed

*\* Abstract: It is very descriptive of the different issues/approaches, but it seems to be missing an obvious conclusion (even if it is a "negative" one).*
Due to the major changes made in the manuscript, the abstract has been rewritten as follow: "China, and especially the North China Plain (NCP), is a highly polluted region. Nevertheless, emission reductions have been occured since about 10 years, starting with $SO_2$ emissions since 2006 and continuing with $NO_x$ emissions since 2010. Recent studies show a decrease in $NO_2$ tropospheric column since 2013 attributed to the $NO_x$ emissions reduction. Quantifying how these emission reductions translates to the ozone concentrations remains unclear due to apparent inconsistencies between surface and satellite observations. In this study, we use the lower tropospheric (LT) columns (surface-6km asl) derived from the IASI-A satellite instrument to describe the variability and trend of LT ozone over the NCP for 2008-2016. First, we investigate the IASI retrieval stability and robustness based on the influence of atmospheric conditions (thermal conditions, aerosol loading) and retrieval sensitivity changes. We compare IASI-A observations with the independent IASI-B instrument aboard the Metop-B satellite as well as surface and ozonesonde measurements. The conclusion of this evaluation is that the LT ozone columns retrieved from IASI-A are reliable to derive trend representative of the lower/free troposphere (3-5 km). Deseasonalized monthly timeseries of LT ozone show two distinct periods: a first period (2008-2012) with no significant trend (< -0.1 %/yr) and a second period (2013-2016) with a highly significant negative trend of -1.2 %/yr, leading to an overall significant trend of -0.77 %/yr for 2008-2016. We explore the dynamical and chemical factors that could explain these negative trends using a multivariate linear regression model and chemistry-transport model simulations to evaluate the sensitivity of ozone to NOx emissions reduction. The results

show that the negative trend observed from IASI for 2013-2016 is almost equally attributed to large-scale dynamical processes and emissions reduction, the large El Nino event in 2015-2016 and the NOx emissions reduction being the main contributors. For the entire period 2008-2016, large-scale dynamical processes explain more than half of the observed trend, with a possible reduction of the stratosphere-to-troposphere exchanges being the main contribution. Large-scale transport and advection evaluated using CO as a proxy contributes for a small part of the trends (~10%). However, a residual significant negative trend remains showing the limitation of linear regression models to account for non-linear processes such as ozone chemistry and stress the need of a detailed evaluation of changes in chemical regimes with the altitude."

*Page 2, line 7: Somewhere in the manuscript there should be more said about the potential discrepancy with MOZAIC data, particularly since the hypothesis is proposed that different regions of the troposphere are NOX or VOC limited. I presume that a lot of the MOZAIC data comes from higher altitudes in the LT, which would also be where IASI retrievals are most sensitive.*
We do not mention the MOZAIS/IAGOS data because China is not well covered by these measurements during the considered period. In the revised version of the manuscript, we stress the need for measurements with a good vertical resolution and refer to the IAGOS program (see the conclusion reproduced above for example).

*Page 4, Section 3.1: Discussion of the time series analysis should have some clar-ifications. References should be given for "Theil-Sen estimator" and "Mann-Kendall test." Use of certain terms, such as "climatological index" are unclear, and should be quantitatively defined through an equation, and not words.*
References have been added and the method used to deseasonalize the timeseries better described (see response to Referee #1)

*Page 5, line 6: "amplitude", not "altitude"* done

*Page 5, lines 11-12, and elsewhere: The authors note the change in O3 in 2013, but it is not clear why this happens. If we do not have a reason it should be stated as an outstanding question.*
One of the hypothesis to the change in 2013 is the response of ozone to the $NO_x$ emissions reduction that starts to be observed using $NO_2$ tropospheric columns since 2013. In the new organization of the corrected manuscript, this assumption is clearly stated. "Intensive emission regulations have been applied in China to reduce $SO_2$ and $NO_x$ emissions during the last years (van der A et al., 2017; Li et al., 2017). The emission reduction of $NO_x$, which are ozone precursors, is observed in the satellite $NO_2$ columns since 2013 as shown in Fig. 8 and reported in very recent inventories (Zheng et al., 2018). On the contrary, the emissions of anthropogenic volatile organic compounds (VOCs) are not regulated and do not show any decrease in the recent years. Zheng et al. (2018) report on an increase between 2010 and 2014 and stagnation since 2014. Stavrakou et al. (Stavrakou et al., 2017) report on an increase in 2013 and 2014 compared to previous years from OMI-HCHO-based emissions and attribute it to the economic recovery after the 2008-2009 crisis. Looking at the timeseries of the HCHO tropospheric columns derived from OMI (De Smedt et al., 2015), available from the TEMIS database, a continuous increase is well observed starting in 2013 and extending to 2016 (Fig. 8). It is worth noting that the increase is less observable in a more recent version of the HCHO product, except for the last year (De Smedt et al., 2018). Thus, one hypothesis is that reductions in surface emissions of $NO_x$ and increase or stagnation in VOCs emissions might cause a decreasing trend in lower tropospheric ozone levels as observed with IASI (Figs. 7-8)."

*Section 3.3, explicative variables: This is a very good discussion. One item that should be discussed in more detail is why the NOx abundances in the LT are not correlated with the O3. I would expect some correlation away from the surface?*

The NO$_x$ abundances show an increase between 2008-2010 followed by a stagnation (2010-2012) and a decrease since 2013 (see Fig. 8 of the revised manuscript, reported in the response to Referee #1). LT ozone shows a rather flat timeseries until 2012/2013 and a decrease since 2013. The non-monotone variations of the NO$_2$ timeseries combined with non-linear chemistry of ozone explain the failure of the linear regression model to correlate the NO$_2$ and O$_3$ timeseries for 2008-2016. Following recommendations of Referees #1 and #2, new analyses have been done. For example, the linear regression model has been applied for 2013-2016 and in that case, NO$_x$ emissions reduction explains about 40% of the negative trend.

*Page 7, line 10: "A period for which strong stratospheric intrusions..."* done

*Figure 4 and Table 1: I found Figure 4 difficult to interpret. The residuals are, of course, all over the place, and impacts are sometimes difficult to see, for example, the impact of PV on residuals at beginning of 2003 (it's only one point!). The general trends become slowly apparent, but the large variations in residuals mask some of the impact of the different explicative variables. I find that Table 1 does a much better job of succinctly summarizing the results, which is harder to get from Figure 4. One suggestion for Table 1, etc., is that the same analysis be carried out for the attribution to the trends in the 2013-2016 period. Given that this period has a large impact on the overall trend, it would be interesting to see how the attributions work for this period.*
We agree that changes from one panel to another were difficult to see in Fig. 4. The new analyses done for the revised version of the manuscript have conducted us to apply the multivariate linear regression model on 3-month rolling averages as we are interested more in explaining the trend than the punctual variations. To draw clearer conclusions the way the analyses of the fitting residual are presented (Table 1) has also been changed. We now present the residual trends of the fitting residual when introducing significant variables one by one in the fit. Following the referee's recommendation, we applied the regression model to the 2013-2016 period with much stronger and clearer conclusions (see response to Referee #1 comments).

*Page 9, lines 5-6. "similar to the trend derived from the data with no filters'?* done

*Page 9, lines 22-25 and Figure 6. What is being plotted in figure 6, i.e., what are the units for the "AK stability" in the y-axis?*
The axis-label was confusing. We changed them to LT columns and added the following title to the figure "Evaluation of the retrieval and AK stability in "no-trend" case".

*Page 10, line 4: Should read: THUS, the comparison of IASI-A and B...*
done

*Page 10, line 15: "The criterion on the time difference has been RELAXED*
done

*Page 12, line 8: "We consider daytime surface observations only ON the days*
done

*Page 12, line 10: "a recent study shows THAT the downward mixing...*
done

**Reorganization of the revised manuscript**
Additional work has been done to address the referee's comments. The new analyses bring new results, which help to improve the discussion and clarify the conclusions of the paper. In order to present the new results and conclusion, major changes have been done in the manuscript. The organization of the manuscript has been changed as follows: Section 1 – introduction, Section 2 – description of IASI data,

the method to calculated the trends, and the developed multivariate linear regression model, Section 3 – evaluation of the instrumental and retrieval stability of IASI and discussion about the reliability of the IASI derived trends, comparison with independent measurements (IASI-B, ozonesondes), Section 4 – analysis of the variability and trends derived from IASI-A, evaluation and discussion about the role of $NO_x$ emissions reduction using surface measurements and CHIMERE simulations, evaluation of the explicative variables and processes of the trend using the multivariate linear regression model, Section 5 – conclusions.

The model simulations (CHIMERE) used in our study has been performed by Mathieu Lachatre and Audrey Fortems-Cheiney. Thus, they are included as co-authors of the study.

---

## Author Response (AR2)

Dear Editor,

All the corrections asked by the referee have been taken into account and included in the revised version of the manuscript.

- P7L21: "with no" has been changed to "without"
- P9L7: "prove" has been changed to "demonstrate"
- P9L20: the sentence has been rephrased: "Almost all of the slopes are not statistically significant"
- P13L17 : the term "rolling average" has been changed to "moving average" all over the text.
- Figure 8 has been modified to include the standard deviation of the moving average. The caption indicates now that the average is done over the NCP.

Thank you in advance for considering the publication of the manuscript in ACP.

Sincerely,

Gaëlle Dufour

[revised manuscript text omitted]